# RECURSIVE TIME SERIES DATA AUGMENTATION

**Amine Mohamed Aboussalah**[1]**, Minjae Kwon**[2]**, Raj G Patel**[3]**, Cheng Chi**[3]**, Chi-Guhn Lee**[3]
[1]New York University, [2]University of Virginia, [3]University of Toronto
`ama10288@nyu.edu`
`hbt9su@virginia.edu`
`{rajg.patel, cheng.chi}@mail.utoronto.ca`
`cglee@mie.utoronto.ca`

## ABSTRACT

Time series observations can be seen as realizations of an underlying dynamical system governed by rules that we typically do not know. For time series learning tasks we create our model using available data. Training on available realizations, where data is limited, often induces severe over-fitting thereby preventing generalization. To address this issue, we introduce a general recursive framework for time series augmentation, which we call the Recursive Interpolation Method (RIM). New augmented time series are generated using a recursive interpolation function from the original time series for use in training. We perform theoretical analysis to characterize the proposed RIM and to guarantee its performance under certain conditions. We apply RIM to diverse synthetic and real-world time series cases to achieve strong performance over non-augmented data on a variety of learning tasks. Our method is also computationally more efficient and leads to better performance when compared to state of the art time series data augmentation.

## 1 INTRODUCTION

The recent success of machine learning (ML) algorithms depends on the availability of a large amount of data and prodigious computing power, which in practice are not always available. In real world applications, it is often impossible to indefinitely sample and ideally, we would like the ML model to make good decisions with a limited number of samples. To overcome these issues, we can exploit additional information such as the structure or invariance in the data that help the ML algorithms efficiently learn and focus on the most important features for solving the task. In ML, the exploitation of structure in the data has been handled using four different yet complementary approaches: 1) Architecture design, 2) Transfer learning, 3) Data representation, and 4) Data augmentation. Our focus in this work is on data augmentation approaches in the context of time series learning.

Time series representations do not expose the full information of the underlying dynamical system Prado (1998) in a way that ML can easily recognize. For instance, in financial time series data, there are patterns at various scales that can be learned to improve performance. At a more fundamental level, time series are one-dimensional projections of a hypersurface of data called the phase space of a dynamical system. This projection results in a loss of information regarding the dynamics of the system. However, we can still make inferences about the dynamical system that projects a time series realization. Our approach is to use these inferences to generate additional time series data from the original realization to build richer representations and improve time series pattern identification resulting in more optimal parameters and reduced variance. We show that our methodology is applicable to a variety of ML algorithms.

Time series learning problems depend on the observed historical data used for training. We often use a set of time series data to train the ML model. Each element in the set can be viewed as a sample derived from the underlying stochastic dynamical system. However, each historical time series data sample is only one particular realization of the underlying stochastic dynamical system in the real

world that we are trying to learn. Our work focuses on problems where available realizations are limited but is not limited to these problems. In fact, our method can be applied to any time series learning task such as stock price prediction where we often have a single realization or problems with numerous realizations such as speech recognition where many audio clips are available for training. Let us consider the stock price prediction problem. The task is to predict or classify the trend of future price. Ideally we want our model to perform well by capturing the stochastic dynamics of stock markets. However, we only train the model using a single time series realization or limited historical realizations. As a result, we do not truly capture the characteristic behaviour of the underlying dynamical system. Using the original training data and hence one or a few realizations of the underlying dynamical system usually induces over-fitting. This is not ideal as we want our model to perform well in the stochastic system instead of just a specific realization of that system.

**Contributions.** The contributions of our work are as follows:

- We present a time series augmentation technique based on recursive interpolation.
- We provide a theoretical analysis of learning improvement for the proposed time series augmentation method:
    - We show that our recursive augmentation allows us to control by how much the augmented time series trajectory deviates from the original time series trajectory (Theorem 3.1) and that there is a natural trade-off that is induced when our augmentation deviates considerably from the original time series (Theorem 3.2).
    - We demonstrate that our learning bound depends on the dimension and properties of the time series, as well as the neural network structure (Theorems 3.3 and 3.4).
    - We believe that this work is the first to offer a theoretical ML framework for time series data augmentation with guarantees for variance reduction in the learned model (Theorem 3.5).
- We empirically demonstrate learning improvements using synthetic data as well as real world time series datasets.

**Outline of the paper.** Section 2 presents the literature review. Section 3 defines the notations, the problem setting, and provides the main theoretical results. Section 4 describes the experimental results, and Section 5 concludes with a summary and a discussion of future work.

## 2 RELATED WORK

**Augmentation for Computer Vision.** In the computer vision context, there are multiple ways to augment image data like cropping, rotation, translation, flipping, noise injection and so on. Among them, the mixup technique proposed in Zhang et al. (2018) is similar to our approach. They train a neural network on convex combinations of pairs of images and their labels. However, just applying a static technique to dynamic time series data is not appropriate. Chen et al. (2020) showed that data augmentation has a similar effect to an averaging operation over the orbits of a certain group of transformation that keep the data distribution invariant.

**Augmentation for Time Series.** There is an exhaustive list of transformations applied to time series that are usually used as data augmentation Wen et al. (2020a). Fawaz et al. (2018) described transformations in the time domain such as time warping and time permutation. There are methods that belong to the magnitude domain such as magnitude warping, Gaussian noise injection, quantization, scaling, and rotation Wen & Keyes (2019). There exists other transformations on time series in the frequency and time-frequency domains that are based on Discrete Fourier Transform (DFT). In this context, they apply transformations in the amplitude and phase spectra of the time series and apply the reverse DFT to generate a new time series signal Gao et al. (2020). Besides the transformations in different domains, there are also more advanced methods, including decomposition-based methods such as the Seasonal and Trend decomposition using Loess (STL) method and its variants Cleveland et al. (1990); Wen et al. (2020b), statistical generative models Kang et al. (2020), and learning-based methods. The learning-based methods can be further divided into embedding space DeVries & Taylor (2017), and deep generative models (DGMs) Esteban et al. (2017); Yoon et al. (2019). These approaches are problem dependent and do not offer theoretical guaranteed learning improvement. In addition, the learning-based methods require large amounts of training data.

**Augmentation for Reinforcement Learning (RL).** Laskin et al. (2020) presented the RL with Augmented Data (RAD) module which can augment most RL algorithms that use image data. They have demonstrated that augmentations such as random translate, random convolutions, crop, patch cutout, amplitude scale, and color jitter can enable RL algorithms to outperform complex advanced methods on standard benchmarks. Kostrikov et al. (2021) presented a data augmentation method that can be applied to conventional model-free reinforcement learning (RL) algorithms, enabling learning directly from pixels without the need for pre-training or auxiliary losses. The inclusion of this augmentation method improves performance substantially, enabling a Soft Actor-Critic agent to reach advanced functioning capability on the DeepMind control suite, outperforming model-based methods and contrastive learning. Laskin et al. (2020) and Kostrikov et al. (2021) show the benefit of RL augmentation using convolutional neural networks (CNNs) for static data but do not handle dynamic data such as time series.

Ideally, we would like to have access to more data that is representative of the underlying dynamics of the system or the regime under which we operate. However, we can not randomly add more data as there is a probability that the added data might not be representative of our stochastic dynamical system. To ensure that we are able to add meaningful data without disturbing the properties of the original data, we introduce a new approach called Recursive Interpolation Method. Our paper proposes a recursive interpolation method for time series as a tool for generating data augmentations.

# 3 THEORETICAL FRAMEWORK FOR RECURSIVE INTERPOLATION METHOD

## 3.1 RECURSIVE INTERPOLATION METHOD (RIM)

In our setting, we consider each time series sample as one realization from the underlying dynamical system. The realization consists of features along the time axis. Let $d + 1$ be the dimension of the time series sample and $\{0, 1, \ldots, k\}$ be the label set for each sample. Then, each sample belongs to $\mathbb{R}^{d+1} \times \{0, 1, \ldots, k\}$. Let $S = \{s_0, s_1, \ldots, s_N\}$ be the collection of the time series samples. Let $\mathcal{D}$ be a distribution with support $[0, 1)$ and $\lambda_i$ be drawn from $\mathcal{D}$ independently denoted by $\lambda_i \sim \mathcal{D}$ for time $i \in [1 : d]$. Let us denote $\vec{\lambda} = (\lambda_1, \ldots, \lambda_d)$ to be the vector of interpolations. For the sake of notational simplicity, in the rest of the paper, we denote $\vec{\lambda} \triangleq \lambda$ and $\lambda \sim \mathcal{D}$ means that each component $\lambda_i$ of $\lambda$ is sampled independently from $\mathcal{D}$.

For a given vector $\lambda \in [0, 1)^d$ and a time series sample $s \in S$, we generate an augmented time series sample $s_\lambda$ as follows. Let us consider an original time series sample $s = (x_0, x_1, \ldots, x_d, y)$ where the time series features $(x_0, x_1, \ldots, x_d) \in \mathbb{R}^{d+1}$ and $y \in \{0, 1, \ldots, k\}$ being the label of the corresponding time series sample. For each $\lambda = (\lambda_1, \ldots, \lambda_d) \in [0, 1)^d$ ($\lambda_0$ is considered to be a dummy value), we define an augmented sample $s_\lambda = (x_{0,\lambda_0}, x_{1,\lambda_1}, \ldots, x_{d,\lambda_d}, y)$ such that

$$x_{i,\lambda_i} = (1 - \lambda_i)x_i + \lambda_i x_{i-1,\lambda_{i-1}} \quad \text{and} \quad x_{0,\lambda_0} = x_0 \tag{1}$$

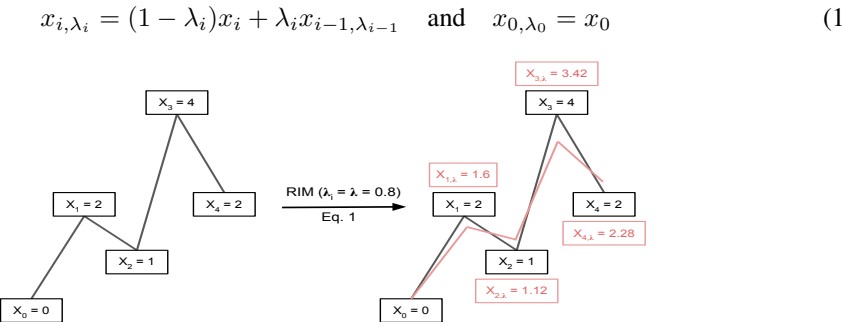

Figure 1: Illustration of RIM.

The newly generated augmented sample $s_\lambda$ has the same label $y$ as the original sample $s$. Our recursive methodology allows us to generate a new time series realization that preserves the trajectory of the original data within some bound (Theorem 3.1).

**Notations.** Let $l(s, \theta)$ be a loss function defined on a sample $s$ and a model parameter $\theta$ and similarly $l(s_\lambda, \theta)$ is the loss defined on the augmented sample using a model parameter $\theta$. We denote that the distribution $\mathcal{P}$ is parametrized by $\theta_*$ on the sample space $\mathcal{S}$. Given few realizations $\{s_i\}_{i \in [0:N]}$ from the distribution $\mathcal{P}$, we set:

$$\theta_* = \underset{\theta}{\operatorname{argmin}} \, \mathbb{E}_{s \sim \mathcal{P}}[l(s, \theta)]$$

$$\hat{\theta} = \underset{\theta}{\operatorname{argmin}} \, \frac{1}{N+1} \sum_{i=0}^{N} l(s_i, \theta)$$

$$\theta_{\text{aug}} = \underset{\theta}{\operatorname{argmin}} \, \mathbb{E}_{s \sim \mathcal{P}}[\mathbb{E}_{\lambda \sim \mathcal{D}}[l(s_\lambda, \theta)]] \tag{2}$$

$$\hat{\theta}_{\text{aug}} = \underset{\theta}{\operatorname{argmin}} \, \frac{1}{N+1} \sum_{i=0}^{N} \mathbb{E}_{\lambda \sim \mathcal{D}}[l(s_{i,\lambda}, \theta)]$$

$$\mathcal{R}_n(l \circ \Theta) = \mathbb{E}_{\epsilon_i \sim \mathcal{E}} \left[ \sup_{\theta \in \Theta} \left| \frac{1}{N+1} \sum_{i=0}^{N} \epsilon_i l(s_i, \theta) \right| \right]$$

Using the notations above, our recursive time series data augmentation minimizes the augmented loss under which we take the expectation over the augmented sample space. We call it average augmented loss and denote it by $l_{\text{aug}}(s, \theta) = \mathbb{E}_{\lambda \sim \mathcal{D}}[l(s_\lambda, \theta)]$.

The three most important aspects of our theoretical framework are the characterization of our recursive time series augmentation: the trade-off that is induced in the learning parameter space (Section 3.2), the learning bound showing the impact of the structural properties of time series and the neural network on the learning parameters (Section 3.3), and better parameter learning with reduced variance when using augmented samples compared to the non augmented ones (Section 3.4).

## 3.2 LEARNING BOUND CONNECTING ORIGINAL TIME SERIES AND AUGMENTED TIME SERIES

We define the recursively interpolated time series and show that the augmented time series samples can deviate from the original ones. We measured this deviation from the original time series using a norm distance between the augmented time series and the original one. We show that this distance is bounded, where the bound depends on the characteristics of the time series features (Theorem 3.1).

Let $\operatorname{sign}(t) = \begin{cases} 0 & \text{if } t = 0 \\ 1 & \text{if } t > 0 \\ -1 & \text{if } t < 0 \end{cases}$, $\delta_{ab} = \begin{cases} 1 & \text{if } a = b \\ 0 & \text{if } a \neq b \end{cases}$, and $\mathcal{D}$ be a distribution with support $[0, 1)$.

**Theorem 3.1.** (Characterization of recursive augmentation) If $\lambda_n \sim \mathcal{D}$ and $g(\lambda_n) = (1 - \lambda_n)(1 - \delta_{0n}) + (1 - \operatorname{sign}(n))$, then the following holds. Let $n \in [0 : N]$.

(1) $x_{n, \lambda_n} = \sum_{k=0}^{n} (\prod_{i=k+1}^{n} \lambda_i) g(\lambda_n) x_n$ where $\lambda_j \sim \mathcal{D}$ for $j \geq 1$ and $\lambda_0$ is a dummy value.

(2) Let $\|\cdot\|$ be a norm, $e = \mathbb{E}[\mathcal{D}]$, $m' = \max_{i \in [1:N]}\{\|x_i - x_{i-1}\|\}$ and $m = \max_{i \in [0:N]}\{\|x_i\|\}$. Then

$$\|\mathbb{E}_{\lambda_1, \ldots, \lambda_n}[(x_{n, \lambda_n} - x_n)]\| \leq \min\{(3e)m, \frac{e}{1-e}m', Nem'\}, \tag{3}$$

$$\mathbb{E}_{\lambda_1, \ldots, \lambda_n}[\|(x_{n, \lambda_n} - x_n)\|] \leq 2\lambda_n m \tag{4}$$

Recall that $s = (x_0, x_1, \ldots, x_d, y)$ and $s_\lambda = (x_{0, \lambda_0}, x_{1, \lambda_1}, \ldots, x_{d, \lambda_d}, y)$, we have

$$\|s - s_\lambda\|_2 = \sqrt{\sum_{i=0}^{d} (x_i - x_{i, \lambda_i})^2} \leq \sum_{i=0}^{d} |x_i - x_{i, \lambda_i}| = \|s - s_\lambda\|_1$$

Note that we used the $l_2$ norm for simplicity but in general, any norm satisfies the inequality. Hence, measuring the distance of each feature between the augmented time series sample and the original one gives us some information about how far they deviate from each other.

**Theorem 3.2.** (Learning bound using characterization of recursive augmentation) Without any loss of generality, let $l(\cdot, \cdot) \in [0,1]^1$ be a loss function with Lipschitz condition and $S = \{s_0, s_1, \ldots, s_N\}$ be the collection of the time series samples. Then with probability at least $1 - \delta$ over the samples $\{s_i\}_{i \in [0:N]}$, we have

$$\mathbb{E}_{s \sim \mathcal{P}}[l(s, \hat{\theta}_{\text{aug}})] - \mathbb{E}_{s \sim \mathcal{P}}[l(s, \theta_*)] < 2\mathcal{R}_N(l_{\text{aug}} \circ \Theta) + \sqrt{\frac{2\log(2/\delta)}{N+1}} + 2L_{\text{Lip}}\mathbb{E}_{s \sim \mathcal{P}}\mathbb{E}_{\lambda \sim \mathcal{D}}[\|s_\lambda - s\|]. \quad (5)$$

Moreover, we have

$$\mathcal{R}_N(l_{\text{aug}} \circ \Theta) \leq \mathcal{R}_N(l \circ \Theta) + \max_{i \in \{0, \ldots, N\}} L_{\text{Lip}}\mathbb{E}_{\lambda \sim \mathcal{D}}[\|s_{i,\lambda} - s_i\|]. \quad (6)$$

with $\mathbb{E}_{\lambda \sim \mathcal{D}}[\|s_{i,\lambda} - s\|] \leq 2mde$, where $e = \mathbb{E}[\mathcal{D}]$, $d + 1$ is the dimension of the time series sample, and $m = \max_{i \in [0:N]}\{\|x_i\|\}$.

Theorem 3.2 represents the standard argument for regret bounds using Rademacher complexity. The term in the left in (Eq.5) is what we call the generalization error or sometimes simply the risk, and the term in the right is called the regret excess risk. Theorem 3.2 tells us how good is our parameter estimation compared to the optimal parameters. Our bound is governed by three terms: 1) Rademacher complexity using the augmented data, 2) the sample size, 3) and the distance between our original time series sample and the augmented one. Now if the distance in (Eq.5) between time series samples before and after augmentation goes to zero then the learning bound is more tight. Another implication of the distance being small is that our recursive augmentation method decreases the Rademacher complexity as demonstrated by (Eq.6). Decreasing Rademacher complexity simply ends up producing a tighter bound on the parameter space.

On the other hand, if the distance in (Eq.5 and Eq.6) becomes large (*i.e.* the augmented time series samples deviate considerably from the original ones), then we can not guarantee with high probability that our method will always outperform the non augmented case. This is a trade-off frequently observed in learning theory.

### 3.3 LEARNING BOUND CONNECTING THE STRUCTURAL PROPERTIES OF TIME SERIES AND NEURAL NETWORKS

**Theorem 3.3.** Let $f_\theta$ be a neural network with a model parameter $\theta$, ReLU activations, and sigmoid function as the activation function for the last layer denoted by $f_\theta(x) = \sigma(g_\theta(x))$ where $g_\theta(x) = \nabla g_\theta^T x + b$ is the pre-activation signal for the last layer. Then the cross entropy loss function $l(\cdot, \cdot)$ has error bound as follows

$$\|l(s_\lambda, \theta) - l(s, \theta)\| \leq \sqrt{d}\left(\|A\|_F + \sum_{i=1}^{d}\|B_i\|_F\right)\|\nabla g_\theta\| \quad (7)$$

where

$$A = \begin{pmatrix} 0 & \vec{0}^T \\ \hline & \frac{\partial x_{1,\lambda_1}}{\partial \lambda_1}\big|_{\lambda=\vec{0}} & \frac{\partial x_{2,\lambda_2}}{\partial \lambda_1}\big|_{\lambda=\vec{0}} & \cdots & \frac{\partial x_{d,\lambda_d}}{\partial \lambda_1}\big|_{\lambda=\vec{0}} \\ & 0 & \frac{\partial x_{2,\lambda_2}}{\partial \lambda_2}\big|_{\lambda=\vec{0}} & \cdots & \frac{\partial x_{d,\lambda_d}}{\partial \lambda_2}\big|_{\lambda=\vec{0}} \\ \vec{0} & \vdots & \vdots & \vdots & \vdots \\ & 0 & 0 & \cdots & \frac{\partial x_{d,\lambda_d}}{\partial \lambda_d}\big|_{\lambda=\vec{0}} \end{pmatrix} \quad B_i = \begin{pmatrix} 0 & \vec{0}^T \\ \hline & \frac{\partial^2 x_{1,\lambda_1}}{\partial \lambda_i \partial \lambda_1}\big|_{\lambda=\vec{0}} & \frac{\partial^2 x_{2,\lambda_2}}{\partial \lambda_i \partial \lambda_1}\big|_{\lambda=\vec{0}} & \cdots & \frac{\partial^2 x_{d,\lambda_d}}{\partial \lambda_i \partial \lambda_1}\big|_{\lambda=\vec{0}} \\ & 0 & \frac{\partial^2 x_{2,\lambda_2}}{\partial \lambda_i \partial \lambda_2}\big|_{\lambda=\vec{0}} & \cdots & \frac{\partial^2 x_{d,\lambda_d}}{\partial \lambda_i \partial \lambda_2}\big|_{\lambda=\vec{0}} \\ \vec{0} & \vdots & \vdots & \vdots & \vdots \\ & 0 & 0 & \cdots & \frac{\partial^2 x_{d,\lambda_d}}{\partial \lambda_i \partial \lambda_d}\big|_{\lambda=\vec{0}}, \end{pmatrix}$$

$\vec{0} \in \mathbb{R}^d$ denoted by column vector, and $\|\cdot\|_F$ is a Frobenius norm.

Theorem 3.3 shows how the structural properties of the time series ($A$ and $B$) and the neural network architecture ($\nabla g_\theta$) can affect the learning process. The interpolation vector $\lambda$ will determine velocity $A$ and accelerations $\{B_i\}_{i \in [1:d]}$ of the features for the augmented sample $s_\lambda$.

---

[1]We can choose the range of the loss function to be in any compact and connected subset of $\mathbb{R}$ under the usual topology.

**Theorem 3.4.** (Learning bound with structural properties on time series and neural network) Let $l$ be the cross entropy loss function such that $l(\cdot, \cdot) \in [0, 1]$. Then with probability at least $1 - \delta$ over the samples $\{s_i\}_{i \in [0:N]}$, we have

$$\mathbb{E}_{s \sim \mathcal{P}}[l(s, \hat{\theta}_{\text{aug}})] - \mathbb{E}_{s \sim \mathcal{P}}[l(s, \theta_*)] < 2\mathcal{R}_N(l_{\text{aug}} \circ \Theta) + \sqrt{\tfrac{2 \log(2/\delta)}{N+1}} + 2\sqrt{d}\left(\mathcal{A} + \textstyle\sum_{i=1}^d \mathcal{B}_i\right)\|\nabla g_\theta\| \quad (8)$$

Moreover, we have

$$\mathcal{R}_N(l_{\text{aug}} \circ \Theta) \leq \mathcal{R}_N(l \circ \Theta) + \sqrt{d}\left(\mathcal{A} + \sum_{i=1}^d \mathcal{B}_i\right)\|\nabla g_\theta\|. \quad (9)$$

Theorem 3.4 reveals that there is a trade off between the dimension of features of a time series sample, the $\lambda$, and the neural network architecture. $\mathcal{A}$ and $\{\mathcal{B}_i\}_{i \in [1:d]}$ are the bounds for the velocity and accelerations, respectively, which are computed using the interpolation vector. The last term of (Eq.8) is constructed by the gradient of the pre-activation $g_\theta$ with respect to an input, the dimension of features of a time series sample, and the bound for the velocity and accelerations induced by the interpolation vector. Theorem 3.4 tells us how good is our parameter estimation compared to the optimal parameters. Our bound is governed by three terms: 1) Rademacher complexity using the augmented data, 2) the sample size, 3) and a term that depends on the dimension of the features of the time series sample $d$, the structural properties of the time series ($\mathcal{A}$ and $\mathcal{B}_i$), and the neural network architecture ($\nabla g_\theta$). Because our RIM method uses a recursive interpolation between two consecutive features, the order of magnitude of A and B do not change drastically, and hence do not blow up the bound. Another implication of the A and B being small is that our recursive augmentation method decreases the Rademacher complexity as demonstrated by (Eq.9). Decreasing Rademacher complexity simply ends up producing a tighter bound on the parameter space.

### 3.4 VARIANCE REDUCTION

Suppose that we observe a set $\{s_0, s_1, \ldots, s_N\}$ of $N + 1$ samples from the underlying sample space $\mathcal{S}$. Using our RIM method, we can augment the observed sample $s_i$ with a distribution $\mathcal{D}$, which results in the set of augmented samples $\{s_{i,\lambda} \mid \lambda \sim \mathcal{D}\}$ for $s_i$. Based on mild assumptions (please refer to Appendix A.5) on the regularity of the loss function and on the underlying sample space, we have the following results.

**Theorem 3.5.** (Asymptotic normality) Assume $\Theta$ is open. Then $\hat{\theta}$ and $\hat{\theta}_{\text{aug}}$ admit the following Bahadur representation;

$$\sqrt{N+1}(\hat{\theta} - \theta_*) = \frac{1}{\sqrt{N+1}} V_{\theta_*}^{-1} \sum_{i=0}^N \nabla l(s_i, \theta_*) + o_{\mathcal{P}}(1)$$

$$\sqrt{N+1}(\hat{\theta}_{\text{aug}} - \theta_*) = \frac{1}{\sqrt{N+1}} V_{\theta_*}^{-1} \sum_{i=0}^N \nabla l_{\text{aug}}(s_i, \theta_*) + o_{\mathcal{P}}(1) \tag{10}$$

Therefore, both $\hat{\theta}$ and $\hat{\theta}_{\text{aug}}$ are asymptotically normal

$$\sqrt{N+1}(\hat{\theta} - \theta_*) \to \mathcal{N}(0, \Sigma_0) \text{ and } \sqrt{N+1}(\hat{\theta}_{\text{aug}} - \theta_*) \to \mathcal{N}(0, \Sigma_{\text{aug}}) \tag{11}$$

where the covariance is given by

$$\Sigma_0 = V_{\theta_*}^{-1} \mathbb{E}_{s \sim \mathcal{P}}[\nabla l(s, \theta_*) \nabla l(s, \theta_*)^T] V_{\theta_*}^{-1}$$

$$\Sigma_{\text{aug}} = \Sigma_0 - \mathbb{E}_{s \sim \mathcal{P}}[XX^T] \tag{12}$$

where $X = \nabla l(s, \theta_*) - \nabla l_{\text{aug}}(s, \theta_*)$. As a consequence, the asymptotic relative efficiency of $\hat{\theta}_{\text{aug}}$ compared to $\hat{\theta}$ is RE $= \frac{tr(\Sigma_0)}{tr(\Sigma_{\text{aug}})} \geq 1$.

(Eq.11) describes the asymptotic behaviour of the learning parameters. (Eq.12) shows that our recursive time series augmentation reduces the variance of the learning parameters.

## 4 EXPERIMENTS

### 4.1 DESIGN

We show our results on time series classification tasks with different time series augmentation methods. We compare the improvements of downstream task performance due to different data augmentation methods by enlarging the training set when we train the classifiers. To benchmark our RIM approach, we compare our results with that achieved by TimeGAN approach Yoon et al. (2019). We specifically select TimeGAN as they are known to preserve the temporal dynamics thereby maintaining the correlation between variables across time. Furthermore, with the flexibility of the unsupervised GAN framework and the control offered by the supervised training in autoregressive models, comparing our results against those by TimeGANs can provide us a rigorous benchmark against a tested state-of-the-art approach.

We use a relatively small training set with a large testing set so that it is more challenging for classifiers to generalize and data augmentations are favorable. Note that we use data augmentation methods on two classes separately as data augmentations should only preserve properties of that particular class. We use different data augmentation methods on these two classes of time series in our training set as follows: RIM methods can be directly applied to time series within each class to generate new time series for that class to enlarge the original training set such that the generated series are close to original series in that class according to Theorem 3.1. For the TimeGAN baseline, we train two TimeGANs separately using time series from each class. Once these two TimeGANs are trained, they are used to generate time series for each class to enlarge the original training set. We consider four tasks: the first two use synthetic datasets generated by solving 1-dimensional ODEs, and the last two use real-world datasets. We compare testing accuracy using the original data, augmented data with RIM, and augmented data with TimeGANs.

### 4.2 RESULTS

For task 1, we consider solutions to ODEs containing exponential functions, and two classes in our binary classification correspond to the two ODEs with different parameters. For task 2, we consider solutions to ODEs with trigonometric functions. In this setting, ODEs can be thought of as generators that generate time series on which we are performing classification, and ODEs with different parameters invoke different dynamical behaviors on their solutions (time series). For each class, we generate multiple solutions using corresponding ODE with different initial values. To make the learning tasks harder, we add random noise to the solution generated by these ODEs.

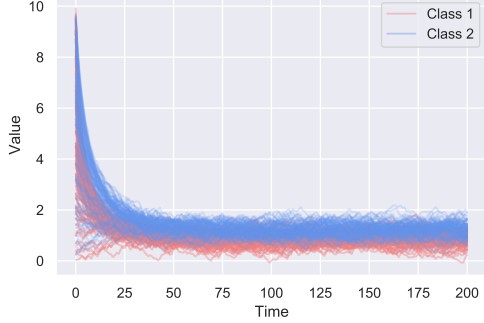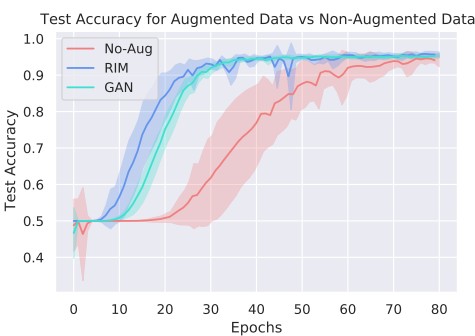

Figure 2: Time series from two classes on the left plot, Test Accuracy for the exponential synthetic ODE system using a Convolutional Neural Network with kernel size=3, filter=32, batch size=16, using BatchNorm and Adam optimizer. The test accuracy plot indicates the resulting mean ± standard deviation from 10 runs.

**Task 1: Synthetic data - Exponential ODEs solutions**

Two ODEs (time series generators) containing exponential functions we use:

class 1 $\frac{dy}{dt} = -0.5y^2 + e^{-y}$; class 2 $\frac{dy}{dt} = -0.3y^2 + 1.5e^{-y}$

**Task 2: Synthetic data - Trigonometric ODEs solutions**
Two ODEs (time series generators) we use:

class 1 $\frac{dy}{dt} = 0.6 + 0.5\sin(y)$; class 2 $\frac{dy}{dt} = 1 + \cos(y)$

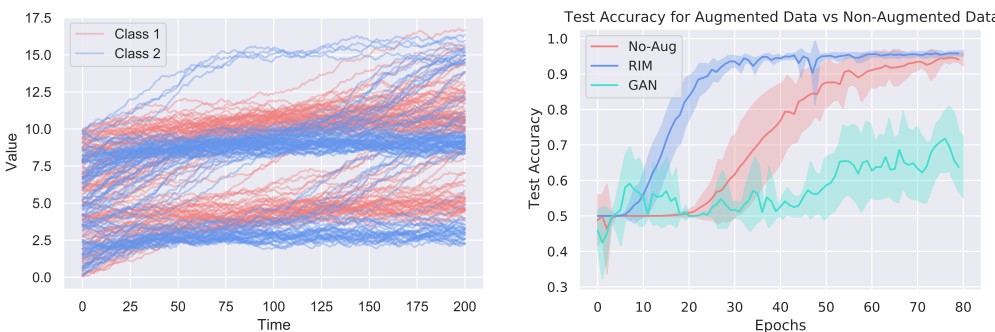

Figure 3: Time series from two classes and Test Accuracy for the trigonometric synthetic ODE system using a Convolutional Neural Network with kernel size=3, filter=32, batch size=16, using BatchNorm and Adam optimizer. The test accuracy plot indicates the resulting mean $\pm$ standard deviation from 10 runs.

**Task 3: Real dataset - Indoor User Movement from the Radio Signal Strength (RSS) Data**
This binary classification task from Bacciu et al. (2014) is associated with predicting the pattern of user movements in real world office environments from time series generated by a Wireless Sensor Network (WSN). The input data contains RSS measured between the nodes of a WSN, comprising of 5 sensors: 4 in the environment and 1 for the user. Data has been collected during movement of the users and labelled to indicate whether the user's trajectory will lead to a change in the room or not. In experiments, we use a subset of the data to form a small training set to challenge our algorithm. We achieve better and more robust test accuracy than the TimeGAN and the non augmented case when using augmented data as reflected in Figure 4. Since $\lambda$ is the only parameter used in our RIM augmentation technique, our ablation study paid very precise attention to the choice of the $\lambda$ parameter. Under this, we tested different $\lambda$ distributions. Given that we were interested in convex combinations between $x_i$ and $x_{i-1,\lambda_{i-1}}$, we had to restrict $\lambda$ between 0 and 1. Two ways to perform this would be: (1) uniformly distribute the weights while sampling $\lambda$; (2) concentrating on a specific part of $\lambda$ distribution. To address (1), we use $\mathcal{U}(0,1)$ which is the main test-bed for all the experiments in the current main text. Whereas to address the (2), we perform studies using beta distribution by varying its shape parameters to focus on specific parts of densities. We tested $Beta(2,2)$, $Beta(0.5,0.5)$ and $Beta(2,5)$, for which the resulting plots can be found in Appendix C. For all these cases, we observed improvements from using RIM compared to non-augmented training, both in terms of a higher final testing accuracy and with fewer training iterations thereby solidifying the effectiveness of RIM.

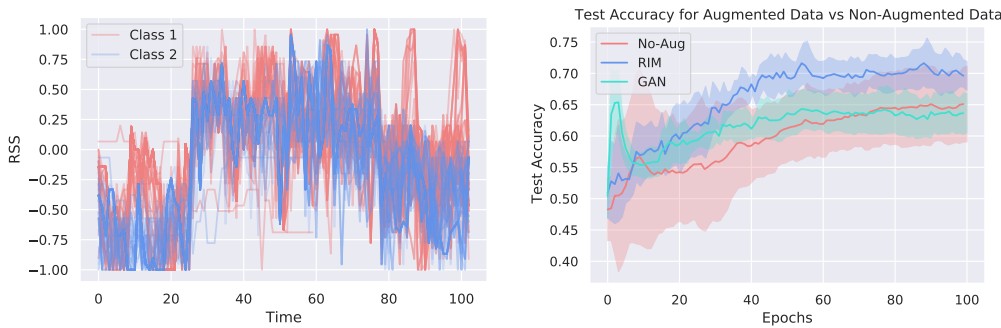

Figure 4: Time series from two classes and Test Accuracy for the Indoor User Movement Classification using a Convolutional Neural Network with kernel size=3, filter=32, batch size=16, using BatchNorm and Adam optimizer. The test accuracy plot indicates the resulting mean $\pm$ standard deviation from 10 runs.

**Task 4: Real dataset - Ford Engine Condition**

We use a subset of the FordA dataset from 2008 WCCI Ford classification challenge Abou-Nasr & Feldkamp (2007). This dataset contains time series corresponding to measurements of engine noise captured by a motor sensor. The goal is to detect the presence of a specific issue with the engine by classifying each time series into issue/no issue classes. We sample 100 time series from FordA to form a small training set to challenge our algorithm and 100 time series for testing. As shown in Figure 5, RIM outperforms the TimeGAN and the non augmented case on the test accuracy.

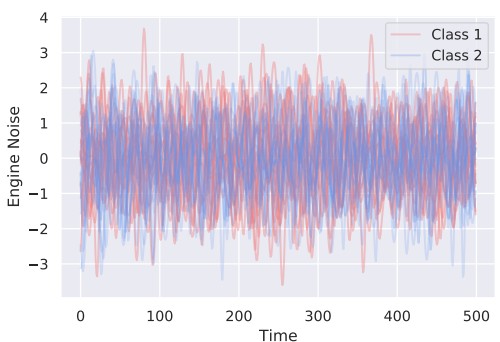 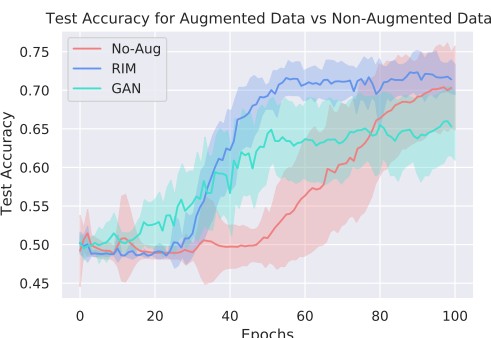

Figure 5: Time series from two classes and Test Accuracy for the Ford Engine Classification with a Convolutional Neural Network with kernel size=3, filter=32, batch size=16, using BatchNorm and Adam optimizer. The test accuracy plot indicates the resulting mean ± standard deviation from 10 runs.

From Figure 2 to Figure 5, we can see that RIM achieves superior performance over the non augmented case. Furthermore, RIM is also able to achieve better or comparable performance than TimeGAN on these tasks without going through the extensive training process associated with GANs. For our experiments, we train the TimeGAN for 2500 epochs (3 hours on Xeon Processors CPU) for synthetic datasets and 5000 epochs (6 hours on Xeon Processors CPU) for real datasets. Visual comparisons of the time series generated by RIM and TimeGAN with the original time series are shown in Appendix B. As expected from Theorem 3.5, RIM has smaller variance and better convergence compared to the non augmented case across all experiments.

## 4.3 EXTENSION TO OTHER LEARNING TASKS

Section 4.2 demonstrates results for time series classification. In this section, we show that RIM can also be used in other learning tasks including continuous time series forecasting and RL. In continuous time series forecasting, we generally have one historical realization and we leverage that to form our training set composed of (x,y) pairs where x is the previous $n$ time steps' data and y is the future step target by decomposing the time series into smaller components. RIM can then be used to generate more time series from the unique realization so that we can enlarge our training set by adding more (x,y) pairs from the original realizations. We also used RIM to augment state trajectories in RL tasks (please refer to pseudo code in Appendix E.2). Preliminary experiments for continuous time series forecasting can be found in the Appendix D and RL tasks can be found in the Appendix E.

## 5 CONCLUSION

We developed a Recursive Interpolation Method (RIM) for time series as a data augmentation technique to learn models accurately with limited data. The RIM is simple, yet effective, supported by theoretical analysis guaranteeing faster convergence. Theoretically, we proved that the RIM guarantees better parameter convergence with reduced variance. Empirically, our methodology outperforms the current state-of-the-art approaches for different real world problem domains and synthetic datasets by obtaining higher accuracy with reduced variance. Because our approach operates on the input time series data, it is invariant to the choice of the ML algorithm. The methodology described in this paper can be used to enhance ML solutions to a wide variety of time series learning problems.

ACKNOWLEDGEMENT

Amine is grateful to Hafida Ines Bouzaouia and Douglas Tweed for their help and constructive comments regarding this work. This research is supported by the Natural Sciences and Engineering Research Council of Canada (NSERC) [grant number 411296449], Canada's federal funding agency for university based research.

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

# Appendix

The code for all the experiments can be found at the following link.

## A    PROOFS

### A.1    PROOF OF THEOREM 3.1

Let $\text{sign}(t) = \begin{cases} 0 & \text{if } t = 0 \\ 1 & \text{if } t > 0 \\ -1 & \text{if } t < 0 \end{cases}$, $\delta_{ab} = \begin{cases} 1 & \text{if } a = b \\ 0 & \text{if } a \neq b \end{cases}$, and $\mathcal{D}$ is a distribution with support $[0, 1)$.

***Proof of Theorem 3.1*** (1) Note that $\lambda_0$ is a dummy value for mathematical convenience and $\prod_{i=k+1}^{n} \lambda_i = 1$ if $n < k + 1$. We prove this by induction on $n$. The case $n = 1$ shows that $x_{1,\lambda_1} = g(\lambda_1)x_1 + \lambda_1 g(\lambda_0)x_0 = (1 - \lambda_1)x_1 + \lambda_1 x_0$ since $g(\lambda_0) = 1$ and $g(\lambda_1) = 1 - \lambda_1$. We now assume that the inequality holds for $n - 1$ and prove it for $n$. By construction of $x_{n,\lambda_n}$,

$$
\begin{aligned}
x_{n,\lambda_n} &= (1 - \lambda_n)x_n + \lambda_n x_{n-1,\lambda_{n-1}} \\
&= (1 - \lambda_n)x_n + \lambda_n \sum_{k=0}^{n-1} \left( \prod_{i=k+1}^{n-1} \lambda_i \right) g(\lambda_k)x_k \\
&= \left( \prod_{i=n+1}^{n} \lambda_i \right) g(\lambda_n)x_n + \sum_{k=0}^{n-1} \left( \prod_{i=k+1}^{n} \lambda_i \right) g(\lambda_k)x_k \\
&= \sum_{k=0}^{n} \left( \prod_{i=k+1}^{n} \lambda_i \right) g(\lambda_k)x_k
\end{aligned}
$$

Thus (1) holds by mathematical induction.

(2) Consider the equation $\|x_{n,\lambda_n} - x_n\|$. Then by (1), the first bound can be found by

$$
\begin{aligned}
\|\mathbb{E}[(x_{n,\lambda_n} - x_n)]\| &= \|\mathbb{E}[\sum_{k=0}^{n} (\prod_{i=k+1}^{n} \lambda_i)g(\lambda_k)x_k - x_n]\| \\
&= \|\mathbb{E}[\sum_{k=0}^{n-1} (\prod_{i=k+1}^{n} \lambda_i)g(\lambda_k)x_k - \lambda_n x_n]\| \\
&= \|\sum_{k=1}^{n-1} e^{n-k}(1 - e)x_k + e^{n-1}x_0 - ex_n\| \\
&\leq \sum_{k=1}^{n-1} e^{n-k}(1 - e)\|x_k\| + e^{n-1}\|x_0\| + e\|x_n\| \\
&\leq (\sum_{k=1}^{n-1} e^{n-k} - \sum_{k=0}^{n-2} e^{n-k} + e^{n-1} + e)m \\
&= (\sum_{k=0}^{n-1} e^{n-k} - \sum_{k=2}^{n-2} e^{n-k} + e)m \leq (e^{n-1} - e^n + 2e)m \leq 3em
\end{aligned}
$$

where $e = \mathbb{E}[\mathcal{D}]$ and $m = \max_{i \in [0:n]} \{\|x_i\|\}$.

Now we prove the second bound. Since

$$
\begin{aligned}
\|x_{n,\lambda_n} - x_n\| &= \|x_n - ((1 - \lambda_n)x_n + \lambda_n x_{n-1,\lambda_{n-1}})\| \\
&= \lambda_n \|x_n - x_{n-1} + x_{n-1} - x_{n-1,\lambda_{n-1}}\| \\
&\leq \lambda_n (\|x_n - x_{n-1}\| + \|x_{n-1} - x_{n-1,\lambda_{n-1}}\|),
\end{aligned}
$$

(13)

By recursively applying (Eq. 13), we obtain

$$\|x_{n,\lambda_n} - x_n\| \leq \sum_{k=1}^{n}(\prod_{i=k}^{n}\lambda_i)\|x_k - x_{k-1}\|. \tag{14}$$

By Jensen's inequality and (Eq. 14), we obtain the second bound

$$\|\mathbb{E}[(x_{n,\lambda_n} - x_n)]\| \leq \mathbb{E}[\|x_{n,\lambda_n} - x_n\|]$$

$$\leq \mathbb{E}[\sum_{k=1}^{n}(\prod_{i=k}^{n}\lambda_i)\|x_k - x_{k-1}\|] \tag{15}$$

$$\leq \sum_{k=1}^{n}e^{n-k+1}m' \leq \min\{\frac{e}{1-e}m', nem'\}$$

where $e = \mathbb{E}[\mathcal{D}]$ and $m' = \max_{i \in [1:n]}\{\|x_i - x_{i-1}\|\}$. Now we show that

$$\mathbb{E}_{\lambda_1,\ldots,\lambda_n}[\|(x_{n,\lambda_n} - x_n)\|] \leq 2\lambda_n m.$$

$$\mathbb{E}_{\lambda_1,\ldots,\lambda_n}[\|(x_{n,\lambda_n} - x_n)\|] = \mathbb{E}[\|\sum_{k=0}^{n-1}(\prod_{i=k+1}^{n}\lambda_i)g(\lambda_k)x_k - \lambda_n x_n\|]$$

$$\leq \mathbb{E}[\sum_{k=0}^{n-1}|(\prod_{i=k+1}^{n}\lambda_i)g(\lambda_k)|\|x_k\| + |\lambda_n|\|x_n\|]$$

$$\leq \mathbb{E}[\sum_{k=0}^{n-1}(\prod_{i=k+1}^{n}\lambda_i)g(\lambda_k) + \lambda_n]m$$

$$= (\lambda_1\lambda_2\cdots\lambda_n + \lambda_2\lambda_3\cdots\lambda_n(1-\lambda_1) + \lambda_3\lambda_4\cdots\lambda_n(1-\lambda_2) + \cdots\lambda_n(1-\lambda_{n-1}) + \lambda_n)m$$

$$= 2\lambda_n m \tag{16}$$

where $m = \max_{i \in [0:N]}\{\|x_i\|\}$. The first inequality holds by Minkowski inequality.

## A.2 PROOF OF THEOREM 3.2

**Neural Networks with ReLU Activations.** Using ReLU activation functions, neural networks are constructed by piecewise linear functions of an input. Such a neural network $g_\theta$ can be formulated by $\nabla g_\theta^T x + b$ where $x$ is an input, $\nabla g_\theta^T$ is the gradient of $g_\theta(x)$ along $x$.

**Lemma A.1.** (Structure of partial derivative) Let $s = (x_0, \ldots, x_d)$ be one sample drawn from time series and $\lambda \in [0,1]^d$. Then

$$\frac{\partial x_{i,\lambda_i}}{\partial \lambda_j} = \begin{cases} 0 & \text{if } i < j \\ (x_{i-1,\lambda_{i-1}} - x_i) & \text{if } i = j \\ (\prod_{k=j+1}^{i}\lambda_k)(x_{j-1,\lambda_{j-1}} - x_j) & \text{if } i > j \end{cases} \tag{17}$$

*Proof.* If $i < j$, then $x_{i,\lambda_i}$ does not depend on $\lambda_j$. Hence $\frac{\partial x_{i,\lambda_i}}{\partial \lambda_j} = 0$.

If $i = j$, then

$$x_{i,\lambda_i} = (1-\lambda_i)x_i + \lambda_i x_{i-1,\lambda_{i-1}} = x_i + \lambda_i(x_{i-1,\lambda_{i-1}} - x_i).$$

Hence $\frac{\partial x_{i,\lambda_i}}{\partial \lambda_j} = (x_{i-1,\lambda_{i-1}} - x_i)$.

If $i > j$, we prove this by induction on $i$. The case $i = j + 1$ shows that

$$x_{j+1,\lambda_{j+1}} = (1-\lambda_{j+1})x_{j+1} + \lambda_{j+1}x_{j,\lambda_j} = x_{j+1} + \lambda_{j+1}(x_{j,\lambda_j} - x_{j+1}).$$

Since $x_{j+1}$ is not dependent on $\lambda_j$ and $\frac{\partial x_{j,\lambda_j}}{\partial \lambda_j} = (x_{j-1,\lambda_{j-1}} - x_j)$, we have $\frac{\partial x_{j+1,\lambda_{j+1}}}{\partial \lambda_j} = \lambda_{j+1}(x_{j-1,\lambda_{j-1}} - x_j)$. We now assume that the inequality holds for $i$ and prove it for $i + 1$. Since

$$x_{i+1,\lambda_{i+1}} = (1-\lambda_{i+1})x_{i+1} + \lambda_{i+1}x_{i,\lambda_i} = x_{i+1} + \lambda_{i+1}(x_{i,\lambda_i} - x_{i+1}) \text{ and}$$

$$\frac{\partial x_{i,\lambda_i}}{\partial \lambda_j} = (\prod_{k=j+1}^{i}\lambda_k)(x_{j-1,\lambda_{j-1}} - x_j),$$

$$\frac{\partial x_{i+1,\lambda_{i+1}}}{\partial \lambda_j} = \lambda_{i+1}\left(\prod_{k=j+1}^{i} \lambda_k\right)(x_{j-1,\lambda_{j-1}} - x_j) = \left(\prod_{k=j+1}^{i+1} \lambda_k\right)(x_{j-1,\lambda_{j-1}} - x_j).$$

As a consequence of mathematical induction, the conclusion holds. $\qquad\square$

We use neural networks with ReLU activations and sigmoid function as the activation function for the last layer. So, a neural network $f_\theta(x) = \sigma(g_\theta(x))$ where $g_\theta(x) = \nabla g_\theta^T x + b$ is the pre-activation signal for the last layer. Recall that we denote by $\vec{\lambda} \triangleq \lambda$ and $\lambda \sim \mathcal{D}$ means that each component $\lambda_i$ of $\lambda$ is sampled independently from $\mathcal{D}$.

***Proof of Theorem 3.2.*** Let $s_\lambda = (x_0, x_{1,\lambda_1}, \ldots, x_{d,\lambda_d}, y)$ and $s$ be one sample. We denote the features by $x_\lambda = (x_0, x_{1,\lambda_1}, \ldots, x_{d,\lambda_d})$ and the label by $y$. Define the matrices

$$A = \begin{pmatrix} 0 & \vec{0}^T \\ \hline & \frac{\partial x_{1,\lambda_1}}{\partial \lambda_1}\big|_{\lambda=\vec{0}} & \frac{\partial x_{2,\lambda_2}}{\partial \lambda_1}\big|_{\lambda=\vec{0}} & \cdots & \frac{\partial x_{d,\lambda_d}}{\partial \lambda_1}\big|_{\lambda=\vec{0}} \\ \vec{0} & 0 & \frac{\partial x_{2,\lambda_2}}{\partial \lambda_2}\big|_{\lambda=\vec{0}} & \cdots & \frac{\partial x_{d,\lambda_d}}{\partial \lambda_2}\big|_{\lambda=\vec{0}} \\ & \vdots & \vdots & \vdots & \vdots \\ & 0 & 0 & \cdots & \frac{\partial x_{d,\lambda_d}}{\partial \lambda_d}\big|_{\lambda=\vec{0}} \end{pmatrix} \tag{18}$$

$$B_i = \begin{pmatrix} 0 & \vec{0}^T \\ \hline & \frac{\partial^2 x_{1,\lambda_1}}{\partial \lambda_i \partial \lambda_1}\big|_{\lambda=\vec{0}} & \frac{\partial^2 x_{2,\lambda_2}}{\partial \lambda_i \partial \lambda_1}\big|_{\lambda=\vec{0}} & \cdots & \frac{\partial^2 x_{d,\lambda_d}}{\partial \lambda_i \partial \lambda_1}\big|_{\lambda=\vec{0}} \\ \vec{0} & 0 & \frac{\partial^2 x_{2,\lambda_2}}{\partial \lambda_i \partial \lambda_2}\big|_{\lambda=\vec{0}} & \cdots & \frac{\partial^2 x_{d,\lambda_d}}{\partial \lambda_i \partial \lambda_2}\big|_{\lambda=\vec{0}} \\ & \vdots & \vdots & \vdots & \vdots \\ & 0 & 0 & \cdots & \frac{\partial^2 x_{d,\lambda_d}}{\partial \lambda_i \partial \lambda_d}\big|_{\lambda=\vec{0}} \end{pmatrix} \tag{19}$$

where $\vec{0} \in \mathbb{R}^d$ denoted by column vector and $e_i$ is a vector whose $i$-th component is 1 and 0 otherwise. Let $l(s_\lambda, \theta) = y\log(f_\theta(x_\lambda)) + (1-y)\log(1 - f_\theta(x_\lambda))$. Denote $l_\theta(\lambda) = l(s_\lambda, \theta)$. Using Taylor expansion of the loss $l$ around $\lambda$, we have

$$l_\theta(\lambda) = l_\theta(\vec{0}) + \sum_{i=1}^{d} \frac{\partial l_\theta(\lambda)}{\partial \lambda_i}\bigg|_{\lambda=\vec{0}} \lambda_i + \frac{1}{2}\sum_{i,j} \frac{\partial^2 l_\theta(\lambda)}{\partial \lambda_i \partial \lambda_j}\bigg|_{\lambda=\vec{0}} \lambda_i \lambda_j + O(\|\lambda\|^2) \tag{20}$$

Note that

$$\begin{aligned} \frac{\partial l_\theta(\lambda)}{\partial x_{i,\lambda_i}} &= y\frac{\partial \log(f_\theta(x_\lambda))}{\partial x_{i,\lambda_i}} + (1-y)\frac{\partial \log(1 - f_\theta(x_\lambda))}{\partial x_{i,\lambda_i}} \\ &= y\frac{\frac{\partial f_\theta(x_\lambda)}{\partial x_{i,\lambda_i}}}{f_\theta(x_\lambda)} - (1-y)\frac{\frac{\partial f_\theta(x_\lambda)}{\partial x_{i,\lambda_i}}}{1 - f_\theta(x_\lambda)} \\ &= \frac{\partial f_\theta(x_\lambda)}{\partial x_{i,\lambda_i}}\frac{y(1 - f_\theta(x_\lambda)) + (y-1)f_\theta(x_\lambda)}{(1 - f_\theta(x_\lambda))f_\theta(x_\lambda)} \\ &= \frac{\partial f_\theta(x_\lambda)}{\partial x_{i,\lambda_i}}\frac{y - f_\theta(x_\lambda)}{(1 - f_\theta(x_\lambda))f_\theta(x_\lambda)} \end{aligned} \tag{21}$$

and

$$\begin{aligned} \frac{\partial f_\theta(x_\lambda)}{\partial x_{i,\lambda_i}} &= \frac{\partial \sigma(g_\theta(x_\lambda))}{\partial x_{i,\lambda_i}} \\ &= \frac{\partial g_\theta(x_\lambda)}{\partial x_{i,\lambda_i}}\sigma(g_\theta(x_\lambda))(1 - \sigma(g_\theta(x_\lambda))) \\ &= \frac{\partial(\nabla g_\theta^T x_\lambda)}{\partial x_{i,\lambda_i}}\sigma(g_\theta(x_\lambda))(1 - \sigma(g_\theta(x_\lambda))) \\ &= \left(\frac{\partial \nabla g_\theta^T}{\partial x_{i,\lambda_i}}x_\lambda + \nabla g_\theta^T \frac{\partial x_\lambda}{\partial x_{i,\lambda_i}}\right)\sigma(g_\theta(x_\lambda))(1 - \sigma(g_\theta(x_\lambda))) \\ &= \nabla g_\theta^T e_i \sigma(g_\theta(x_\lambda))(1 - \sigma(g_\theta(x_\lambda))). \end{aligned} \tag{22}$$

Since the $i$-th feature $x_{i,\lambda_i}$ of $x_\lambda$ depends on $\{\lambda_1, \ldots, \lambda_i\}$, we have

$$
\begin{aligned}
\frac{\partial l_\theta(\lambda)}{\partial \lambda_j} &= \sum_{i=1}^{d} \frac{\partial x_{i,\lambda_i}}{\partial \lambda_j} \frac{\partial l_\theta(\lambda)}{\partial x_{i,\lambda_i}} \\
&= \sum_{i=j}^{d} \frac{\partial x_{i,\lambda_i}}{\partial \lambda_j} \frac{\partial f_\theta(x_\lambda)}{\partial x_{i,\lambda_i}} \frac{y - f_\theta(x_\lambda)}{(1 - f_\theta(x_\lambda))f_\theta(x_\lambda)} \\
&= \sum_{i=j}^{d} \frac{\partial x_{i,\lambda_i}}{\partial \lambda_j} \nabla g_\theta^T e_i \sigma(g_\theta(x_\lambda))(1 - \sigma(g_\theta(x_\lambda))) \frac{y - f_\theta(x_\lambda)}{(1 - f_\theta(x_\lambda))f_\theta(x_\lambda)} \\
&= \sum_{i=j}^{d} \frac{\partial x_{i,\lambda_i}}{\partial \lambda_j} \nabla g_\theta^T e_i (y - f_\theta(x_\lambda))
\end{aligned}
\tag{23}
$$

Thus we have

$$
\sum_{j=1}^{d} \frac{\partial l_\theta(\lambda)}{\partial \lambda_j}\bigg|_{\lambda=\vec{0}} \lambda_j = (y - f_\theta(x))(0, \lambda)^T A \nabla g_\theta
\tag{24}
$$

Hence we have

$$
\left| \sum_{j=1}^{d} \frac{\partial l_\theta(\lambda)}{\partial \lambda_j}\bigg|_{\lambda=\vec{0}} \lambda_j \right| \le \sqrt{d} \|A\|_F \|\nabla g_\theta\|.
\tag{25}
$$

Note that $\|y - f_\theta(x)\| \le 1$.
Now we consider the second partial derivative of the loss function $l_\theta(\lambda)$.

$$
\begin{aligned}
\frac{\partial^2 l_\theta(\lambda)}{\partial \lambda_u \partial \lambda_j} &= \frac{\partial}{\partial \lambda_u}\Big( \sum_{i=j}^{d} \frac{\partial x_{i,\lambda_i}}{\partial \lambda_j} \nabla g_\theta^T e_i (y - f_\theta(x_\lambda)) \Big) \\
&= \sum_{i=j}^{d} \nabla g_\theta^T e_i \bigg( \frac{\partial^2 x_{i,\lambda_i}}{\partial \lambda_u \partial \lambda_j}(y - f_\theta(x_\lambda)) - \\
&\quad \frac{\partial x_{i,\lambda_i}}{\partial \lambda_j} \sum_{k=u}^{d} \frac{\partial x_{k,\lambda_k}}{\partial \lambda_u} \nabla g_\theta^T e_k f_\theta(x_\lambda)(1 - f_\theta(x_\lambda)) \bigg)
\end{aligned}
\tag{26}
$$

We will put (Eq. 26) into (Eq. 20) to calculate the Taylor loss explicitly. For the first term, we have

$$
\begin{aligned}
\sum_{i=1}^{d} \lambda_i \sum_{j=1}^{d} \lambda_j \sum_{l=j}^{d} \nabla g_\theta^T e_l \Big( \frac{\partial^2 x_{l,\lambda_l}}{\partial \lambda_i \partial \lambda_j}(y - f_\theta(x_\lambda)) \Big)\bigg|_{\lambda=\vec{0}} = \\
\sum_{i=1}^{d} \lambda_i (y - f_\theta(x))(0, \lambda)^T B_i \nabla g_\theta
\end{aligned}
\tag{27}
$$

and for the second term, we have

$$
\begin{aligned}
&\sum_{j=1}^{d} \lambda_j \sum_{i=j}^{d} \nabla g_\theta^T e_i \frac{\partial x_{i,\lambda_i}}{\partial \lambda_j} \sum_{l=1}^{d} \lambda_l \sum_{k=l}^{d} \frac{\partial x_{k,\lambda_k}}{\partial \lambda_l} \nabla g_\theta^T e_k f_\theta(x_\lambda)(1 - f_\theta(x_\lambda))\bigg|_{\lambda=\vec{0}} \\
&= \sum_{j=1}^{d} \lambda_j \sum_{i=j}^{d} \nabla g_\theta^T e_i \frac{\partial x_{i,\lambda_i}}{\partial \lambda_j} f_\theta(x)(1 - f_\theta(x))(0, \lambda)^T A \nabla g_\theta \\
&= f_\theta(x)(1 - f_\theta(x))(0, \lambda)^T A \nabla g_\theta \nabla g_\theta^T A^T (0, \lambda).
\end{aligned}
\tag{28}
$$

By combining (Eq. 27 and 28), we have

$$
\sum_{i=1}^{d}\sum_{j=1}^{d}\lambda_i\lambda_j\frac{\partial^2 l_\theta(\lambda)}{\partial\lambda_i\partial\lambda_j}\bigg|_{\lambda=\vec{0}} = \bigg(\sum_{i=1}^{d}\lambda_i(y-f_\theta(x))(0,\lambda)^T B_i\nabla g_\theta\bigg)
$$
$$
-f_\theta(x)(1-f_\theta(x))(0,\lambda)^T A\nabla g_\theta\nabla g_\theta^T A^T(0,\lambda) \tag{29}
$$
$$
\leq \sum_{i=1}^{d}\sqrt{d}\|B_i\|_F\|\nabla g_\theta\|.
$$

The last inequality holds due to

$$
f_\theta(x)(1-f_\theta(x))(0,\lambda)^T A\nabla g_\theta(x)\nabla g_\theta(x)^T A^T(0,\lambda) \geq 0 \tag{30}
$$

By (Eq. 20, 25 and 29), the conclusion holds. $\qquad\square$

### A.3 PROOF OF THEOREM 3.2

We start with a basic inequality frequently used in the proofs.

**Lemma A.2.** (Supremum inequality) Let $f$ and $g$ be functions which have the same domain and range. Then

$$
\sup_\theta|f(\theta)| - \sup_\theta|g(\theta)| \leq \sup_\theta|f(\theta)-g(\theta)| \tag{31}
$$

*Proof.*

$$
\sup_\theta|f(\theta)| = \sup_\theta|f(\theta)-g(\theta)+g(\theta)|
$$
$$
\leq \sup_\theta(|f(\theta)-g(\theta)|+|g(\theta)|)
$$
$$
\leq \sup_\theta|f(\theta)-g(\theta)|+\sup_\theta|g(\theta)|
$$

Thus the conclusion holds. $\qquad\square$

**Notation.** We assume that the distribution $\mathcal{P}$ is parametrized by $\theta_*$ on the sample space $\mathcal{S}$. Let $\{s_i\}_{i\in[0:N]}$ be the collection of samples from the distribution $\mathcal{P}$. We set:

$$
\theta_* = \underset{\theta}{\operatorname{argmin}}\,\mathbb{E}_{s\sim\mathcal{P}}[l(s,\theta)]
$$
$$
\hat{\theta} = \underset{\theta}{\operatorname{argmin}}\,\frac{1}{N+1}\sum_{i=0}^{N}l(s_i,\theta)
$$
$$
\theta_{\mathrm{aug}} = \underset{\theta}{\operatorname{argmin}}\,\mathbb{E}_{s\sim\mathcal{P}}[\mathbb{E}_{\lambda\sim\mathcal{D}}[l(s_\lambda,\theta)]]
$$
$$
\hat{\theta}_{\mathrm{aug}} = \underset{\theta}{\operatorname{argmin}}\,\frac{1}{N+1}\sum_{i=0}^{N}\mathbb{E}_{\lambda\sim\mathcal{D}}[l(s_{i,\lambda},\theta)] \tag{32}
$$
$$
\mathcal{R}_n(l\circ\Theta) = \mathbb{E}_{\epsilon_i\sim\mathcal{E}}\left[\sup_{\theta\in\Theta}\left|\frac{1}{N+1}\sum_{i=0}^{N}\epsilon_i l(s_i,\theta)\right|\right]
$$
$$
l_{\mathrm{aug}}(s,\theta) = \mathbb{E}_{\lambda\sim\mathcal{D}}[l(s_\lambda,\theta)]
$$

**Assumption 1.** Assume that the loss function $l$ satifies Lipschitz condition with respect to the norm.

*Proof of Theorem 3.2.*

$$
\mathbb{E}_{s\sim\mathcal{P}}[l(s,\hat{\theta}_{\mathrm{aug}})] - \mathbb{E}_{s\sim\mathcal{P}}[l(s,\theta_*)] = u_1 + u_2 + u_3 + u_4 + u_5
$$

where

$$u_1 = \mathbb{E}_{s\sim\mathcal{P}}[l(s,\hat{\theta}_{\text{aug}})] - \mathbb{E}_{s\sim\mathcal{P}}[\mathbb{E}_{\lambda\sim\mathcal{D}}[l(s_\lambda,\hat{\theta}_{\text{aug}})]]$$

$$u_2 = \mathbb{E}_{s\sim\mathcal{P}}[\mathbb{E}_{\lambda\sim\mathcal{D}}[l(s_\lambda,\hat{\theta}_{\text{aug}})]]-$$

$$\frac{1}{N+1}\sum_{i=0}^{N}\mathbb{E}_{\lambda\sim\mathcal{D}}[l(s_{i,\lambda},\hat{\theta}_{\text{aug}})]$$

$$u_3 = \frac{1}{N+1}\sum_{i=0}^{N}\mathbb{E}_{\lambda\sim\mathcal{D}}[l(s_{i,\lambda},\hat{\theta}_{\text{aug}})]-$$

$$\frac{1}{N+1}\sum_{i=0}^{N}\mathbb{E}_{\lambda\sim\mathcal{D}}[l(s_{i,\lambda},\theta_*)]$$

$$u_4 = \frac{1}{N+1}\sum_{i=0}^{N}\mathbb{E}_{\lambda\sim\mathcal{D}}[l(s_{i,\lambda},\theta_*)] - \mathbb{E}_{s\sim\mathcal{P}}[\mathbb{E}_{\lambda\sim\mathcal{D}}[l(s_\lambda,\theta_*)]]$$

$$u_5 = \mathbb{E}_{s\sim\mathcal{P}}[\mathbb{E}_{\lambda\sim\mathcal{D}}[l(s_\lambda,\theta_*)]] - \mathbb{E}_{s\sim\mathcal{P}}[l(s,\theta_*)]$$

(33)

We get

$$u_1 + u_5 \le 2\sup_{\theta\in\Theta}\left|\mathbb{E}_{s\sim\mathcal{P}}[l(s,\theta)] - \mathbb{E}_{s\sim\mathcal{P}}[\mathbb{E}_{\lambda\sim D}[l(s_\lambda,\theta)]]\right| \tag{34}$$

where we have

$$\begin{aligned}
\mathbb{E}_{s\sim\mathcal{P}}[l(s,\theta)] - \mathbb{E}_{s\sim\mathcal{P}}[\mathbb{E}_{\lambda\sim D}[l(s_\lambda,\theta)]] &= \mathbb{E}_{s\sim\mathcal{P}}[l(s,\theta) - \mathbb{E}_{\lambda\sim D}[l(s_\lambda,\theta)]] \\
&= \mathbb{E}_{s\sim\mathcal{P}}[\mathbb{E}_{\lambda\sim\mathcal{D}}[l(s,\theta) - l(s_\lambda,\theta)]] \\
&\le L_{\text{Lip}}\mathbb{E}_{s\sim\mathcal{P}}\mathbb{E}_{\lambda\sim\mathcal{D}}[\|s_\lambda - s\|].
\end{aligned} \tag{35}$$

Hence, from (Eq. 34 and 35)

$$u_1 + u_5 \le 2L_{\text{Lip}}\mathbb{E}_{s\sim\mathcal{P}}\mathbb{E}_{\lambda\sim\mathcal{D}}[\|s_\lambda - s\|]. \tag{36}$$

By McDiarmid's inequality, in terms of the probability

$$\mathbb{P}\left(\frac{1}{N+1}\sum_{i=0}^{N}\mathbb{E}_{\lambda\sim\mathcal{D}}[l(s_{i,\lambda},\theta_*)] - \mathbb{E}_{s\sim\mathcal{P}}[\mathbb{E}_{\lambda\sim\mathcal{D}}[l(s_\lambda,\theta_*)]] \ge t\right) \le \exp\left(-\frac{2t^2}{\sum_{i=0}^{N}(\frac{1}{N+1})^2}\right) \tag{37}$$

$u_4$ has the following bound with probability at least $1-\delta$

$$u_4 < \sqrt{\frac{\log(1/\delta)}{2(N+1)}}. \tag{38}$$

Moreover, Rademacher complexity holds for $u_2$, so we have

$$u_2 \le 2\mathcal{R}_N(l_{\text{aug}}\circ\Theta) + 4\sqrt{\frac{2\log(4/\delta)}{N+1}} \tag{39}$$

with probability at least $1-\delta$. By (Eq. 38 and 39), we get the following inequality

$$u_2 + u_4 \le 2\mathcal{R}_N(l_{\text{aug}}\circ\Theta) + 5\sqrt{\frac{2\log(4/\delta)}{N+1}} \tag{40}$$

with probability at least $1-\delta$ where

$$\mathcal{R}_N(l\circ\Theta) = \mathbb{E}_{\epsilon_i\sim\mathcal{E}}\left[\sup_{\theta\in\Theta}\left|\frac{1}{N+1}\sum_{i=0}^{N}\epsilon_i\mathbb{E}_{\lambda\sim\mathcal{D}}[l(s_{i,\lambda},\theta)]\right|\right].$$

where $\epsilon_i$ is a Radamacher variable for all $i\in[N]$.

Since $\hat{\theta}_{\text{aug}}$ is an optimal parameter for $\frac{1}{N+1}\sum_{i=0}^{N}\mathbb{E}_{\lambda\sim\mathcal{D}}[l(s_{i,\lambda},\theta)]$, then

$$u_3 \le 0. \tag{41}$$

From (Eq. 47, 40 and 41), we conclude that

$$\mathbb{E}_{s\sim\mathcal{P}}[l(s,\hat{\theta}_{\text{aug}})]-\mathbb{E}_{s\sim\mathcal{P}}[l(s,\theta_*)] < 2\mathcal{R}_N(l_{\text{aug}}\circ\Theta)+5\sqrt{\frac{2\log(4/\delta)}{N+1}}+2L_{\text{Lip}}\mathbb{E}_{s\sim\mathcal{P}}\mathbb{E}_{\lambda\sim\mathcal{D}}[\|s_\lambda-s\|]. \tag{42}$$

Now we will prove that

$$\mathcal{R}_N(l_{\text{aug}}\circ\Theta) \le \mathcal{R}_N(l\circ\Theta) + \max_{i\in\{0,\dots,N\}} L_{\text{Lip}}\mathbb{E}_{\lambda\sim\mathcal{D}}[\|s_{i,\lambda}-s_i\|]. \tag{43}$$

Since

$$
\begin{aligned}
\mathcal{R}_N(l_{\text{aug}}\circ\Theta)-\mathcal{R}_N(l\circ\Theta) &= \mathbb{E}_{\epsilon_i\sim\mathcal{E}}[\sup_{\theta\in\Theta}\Big|\frac{1}{N+1}\sum_{i=0}^{N}\epsilon_i l_{\text{aug}}(s_i,\theta)\Big| - \sup_{\theta\in\Theta}\Big|\frac{1}{N+1}\sum_{i=0}^{N}\epsilon_i l(s_i,\theta)\Big|] \\
&\le \mathbb{E}_{\epsilon_i\sim\mathcal{E}}[\sup_{\theta\in\Theta}\Big|\frac{1}{N+1}\sum_{i=0}^{N}\epsilon_i l_{\text{aug}}(s_i,\theta) - \frac{1}{N+1}\sum_{i=0}^{N}\epsilon_i l(s_i,\theta)\Big|] \\
&= \mathbb{E}_{\epsilon_i\sim\mathcal{E}}[\sup_{\theta\in\Theta}\Big|\frac{1}{N+1}\sum_{i=0}^{N}\epsilon_i(l_{\text{aug}}(s_i,\theta)-l(s_i,\theta))\Big|] \\
&\le \mathbb{E}_{\epsilon_i\sim\mathcal{E}}[\sup_{\theta\in\Theta}\frac{1}{N+1}\sum_{i=0}^{N}\Big|\epsilon_i(l_{\text{aug}}(s_i,\theta)-l(s_i,\theta))\Big|] \\
&\le \sup_{\theta\in\Theta}\frac{1}{N+1}\sum_{i=0}^{N}\Big|(l_{\text{aug}}(s_i,\theta)-l(s_i,\theta))\Big| \\
&= \sup_{\theta\in\Theta}\frac{1}{N+1}\sum_{i=0}^{N}\Big|\mathbb{E}_{\lambda\sim\mathcal{D}}[l(s_{i,\lambda},\theta)-l(s_i,\theta)]\Big| \\
&\le \max_{i\in\{0,\dots,N\}} L_{\text{Lip}}\mathbb{E}_{\lambda\sim\mathcal{D}}[\|s_{i,\lambda}-s_i\|].
\end{aligned}
\tag{44}
$$

We will show $\mathbb{E}_{\lambda\sim\mathcal{D}}[\|s_{i,\lambda}-s\|] \le 2dem$. Let each sample $s$ (or $s_i$) $\in \mathbb{R}^{d+1}\times\{0,1,\dots,k\}$. Then $s = (x_0,x_1,\dots,x_d,y)$. By the augmentation method, recursively applying convex combinations, we have $s_\lambda = (x_0,x_{1,\lambda_1},x_{2,\lambda_2},\dots,x_{d,\lambda_d},y)$. By Theorem 3.1 (Eq. 4), each $\mathbb{E}_{\lambda_1,\dots,\lambda_j}[\|(x_{j,\lambda_j}-x_j)\|] \le 2\lambda_j m$. Hence

$$
\begin{aligned}
\mathbb{E}_{\lambda\sim\mathcal{D}}[\|s_{i,\lambda}-s\|] &\le \sum_{j=1}^{d}\mathbb{E}_{\lambda\sim\mathcal{D}}[\|x_{\lambda_j,j}-x_j\|] \\
&\le 2\mathbb{E}_{\lambda\sim\mathcal{D}}[(\lambda_1+\lambda_2+\dots+\lambda_d)]m \\
&\le 2mde
\end{aligned}
\tag{45}
$$

where $e = \mathbb{E}[\mathcal{D}]$, $d+1$ is the dimension of the time series sample, and $m = \max_{i\in[0:N]}\{\|x_i\|\}$.

$\square$

## A.4   PROOF OF THEOREM 3.4

Let $\mathcal{A} = \sup_{s\in S}\|A\|_F$ and $\mathcal{B}_i = \sup_{s\in S}\|B_i\|_F$. Then for each $s\sim\mathcal{P}$, we have:

$$\|A\|_F \le \mathcal{A} \quad \text{and} \quad \|B_i\|_F \le \mathcal{B}_i,$$

where $A$ and $B_i$ depend on $s$ for $i\in[d]$. Please refer to (Eq. 18) and (Eq. 19).

***Proof of Theorem 3.4.*** The proof is the almost same with the proof of Theorem 3.2. We only describe the part should be replaced in order to prove Theorem 3.4. Similarly, We start with the

decomposition of the equation as follows

$$\mathbb{E}_{s\sim\mathcal{P}}[l(s,\hat{\theta}_{\mathrm{aug}})] - \mathbb{E}_{s\sim\mathcal{P}}[l(s,\theta_*)] = u_1 + u_2 + u_3 + u_4 + u_5$$

where $u_1, u_2, u_3, u_4$ and $u_5$ are defined in (Eq. 33).
(Eq. 35) in the proof of Theorem 3.2 will be replaced with the following

$$\begin{aligned}
\mathbb{E}_{s\sim\mathcal{P}}[l(s,\theta)] - \mathbb{E}_{s\sim\mathcal{P}}[\mathbb{E}_{\lambda\sim D}[l(s_\lambda,\theta)]] &= \mathbb{E}_{s\sim\mathcal{P}}[l(s,\theta) - \mathbb{E}_{\lambda\sim D}[l(s_\lambda,\theta)]] \\
&= \mathbb{E}_{s\sim\mathcal{P}}[\mathbb{E}_{\lambda\sim\mathcal{D}}[l(s,\theta) - l(s_\lambda,\theta)]] \\
&\leq \sqrt{d}\left(\mathcal{A} + \sum_{i=1}^{d}\mathcal{B}_i\right)\|\nabla g_\theta\|.
\end{aligned} \tag{46}$$

Hence, from (Eq. 34) and (Eq. 46), we have

$$u_1 + u_5 \leq 2\sqrt{d}\left(\mathcal{A} + \sum_{i=1}^{d}\mathcal{B}_i\right)\|\nabla g_\theta\|. \tag{47}$$

The last two lines of (Eq. 44) will be replaced with the following

$$\sup_{\theta\in\Theta}\frac{1}{N+1}\sum_{i=0}^{N}\left|\mathbb{E}_{\lambda\sim\mathcal{D}}[l(s_{i,\lambda},\theta) - l(s_i,\theta)]\right| \leq \sqrt{d}\left(\mathcal{A} + \sum_{i=1}^{d}\mathcal{B}_i\right)\|\nabla g_\theta\| \tag{48}$$

Theorem 3.3 guarantees the inequality. Thus the conclusion holds. $\qquad\square$

### A.5 Proof of Theorem 3.5

**Asymptotic Results.** Note that the sample space is contained in $\mathbb{R}^{d+1} \times \{0,\ldots,k\}$ and the parameter space $\Theta \subseteq \mathbb{R}^p$ where $\{0,\ldots,k\}$ is the label set. Under regularity conditions, it is well known that $\hat{\theta}$ is asymptotically normal with covariance given by the inverse Fisher information matrix. We will see that $\hat{\theta}_{\mathrm{aug}}$ is also asymptotically normal with the covariance. Suppose that we observe a set $\{s_0, s_1, \ldots, s_N\}$ of $N+1$ samples from the underlying sample space $\mathcal{S}$. Using our RIM method, we can augment the observed sample $s_i$ with a distribution $\mathcal{D}$, which results in the set of augmented samples $\{s_{i,\lambda} \mid \lambda\sim\mathcal{D}\}$ for $s_i$. We then decompose $\cup_{i=0}^N\{s_{i,\lambda} \mid \lambda\sim\mathcal{D}\}$ into disjoint union of some sets $S_i$ such that $s_i \in S_i$.

**Assumption 2.** (Disjointness) the sample space $\mathcal{S}$ is the disjoint countable union of all possible augmented sample spaces.

Consider the probability space is $(\mathcal{S}, \mathcal{F}, \mathcal{P})$ where $\mathcal{F}$ is a sigma algebra and $\mathcal{P}$ the corresponding probability measure. Let $\mu$ be a measurable function from $\mathcal{S}$ to $\mathbb{R}^p$ for some $p \in \mathbb{N}$. For each sample $s = (x, y) \in \mathcal{S}$, define $\overline{\mu}(s) = \mathbb{E}[\mu \mid S_i]$ where $s \in S_i$. Let's assume that we observe $N+1$ samples $\{s_0, s_1, \ldots, s_N\}$ from the underlying sample space. Then by Assumption 2, the underlying sample space is decomposed into $\cup_{i=1}^\infty S_i$. i,e $\mathcal{S} = \cup_{i=1}^\infty S_i$. This is well-defined by Assumption 2, and $\overline{\mu}$ is a measurable function. Let $\mu_i$ be the expectation value of $\mu$ over $S_i$.

**Lemma A.3.** (Effects of the average function) With notation as above, the following holds.

1. The law of total expectation: $\mathbb{E}_{s\sim\mathcal{P}}[\mu] = \mathbb{E}_{s\sim\mathcal{P}}[\overline{\mu}]$.

2. The law of total covariance: $\mathrm{Cov}_{s\sim\mathcal{P}}\mu = \mathbb{E}_{s\sim\mathcal{P}}[\mathrm{Cov}(\mu|\overline{\mu})] + \mathrm{Cov}_{s\sim\mathcal{P}}\overline{\mu}$.

*Proof.* From the disjointedness of the underlying sample space, for each $s \in \mathcal{S}$, we have $s \in S_i$ for some $i \in \mathbb{N}$

$$\mathbb{E}_{s\sim\mathcal{P}}[\overline{\mu}] = \mathbb{E}_{s\sim\mathcal{P}}[\mathbb{E}_{s\sim\mathcal{P}}[\mu \mid S_i]] = \mathbb{E}_{s\sim\mathcal{P}}[\mathbb{E}_{s\sim\mathcal{P}}[\mu \mid \overline{\mu} = \mu_i]] = \mathbb{E}_{s\sim\mathcal{P}}[\mu]. \tag{49}$$

Thus the law of total expectation and the law of total covariance naturally follow. $\qquad\square$

Under mild assumptions for a given loss function, we show that the average loss function $l_{\mathrm{aug}}$ inherits the same properties from the non augmented loss function, where $l_{\mathrm{aug}}(s,\theta) = \mathbb{E}_{\lambda\sim\mathcal{D}}[l(s_\lambda,\theta)]$.

**Assumption 3.** (Regularity of the loss function) For the loss function $l(\cdot, \theta)$, we assume that

1. For the minimizer $\theta_*$ of the population risk and any $\epsilon > 0$, we have

$$\sup_{\{\|\theta - \theta_*\| \geq \epsilon \,|\, \theta \in \Theta\}} \mathbb{E}_{s \sim \mathcal{P}}[l(s, \theta)] > \mathbb{E}_{s \sim \mathcal{P}}[l(s, \theta_*)]$$

2. For every $\epsilon > 0$, there exists a function $l' \in L^2(\mathcal{P})$ such that for almost every $s$ and for every $\theta_1, \theta_2 \in N(\theta_0, \epsilon)$, we have

$$|l(s, \theta_1) - l(s, \theta_2)| \leq l'(s)\|\theta_1 - \theta_2\|$$

3. Uniform weak law of large number holds

$$\sup_{\theta \in \Theta} \left| \frac{1}{N+1} \sum_{i=0}^{N} l(s_i, \theta) - \mathbb{E}_{s \sim \mathcal{P}}[l(s, \theta)] \right| \to 0$$

4. For each $\theta$ in $\Theta$, the map $s \to l(s, \theta)$ is measurable

5. The map $\theta \to l(s, \theta)$ is differentiable at $\theta_*$ for almost every $s$

6. The map $\theta \to \mathbb{E}_{s \sim \mathcal{P}}[l(s, \theta)]$ admits a second-order Taylor expansion at $\theta_*$ with non-singular second derivatives matrix $V_{\theta_*}$

**Proposition A.4.** For the pair $(\theta_{\text{aug}}, l_{\text{aug}})$, the following property holds. For every $\epsilon > 0$, there exists a function $l'_{\text{aug}} \in L^2(\mathcal{P})$ such that for almost every $s$ and for every $\theta_1, \theta_2 \in N(\theta_0, \epsilon)$, we have

$$|l_{\text{aug}}(s, \theta_1) - l_{\text{aug}}(s, \theta_2)| \leq l'_{\text{aug}}(s)\|\theta_1 - \theta_2\|$$

*Proof.* Since we have

$$
\begin{aligned}
|l_{\text{aug}}(s, \theta_1) - l_{\text{aug}}(s, \theta_2)| &= |\mathbb{E}_{\lambda \sim \mathcal{D}}[l(s_\lambda, \theta_1) - l(s_\lambda, \theta_2)]| \\
&\leq \mathbb{E}_{\lambda \sim \mathcal{D}}[|l(s_\lambda, \theta_1) - l(s_\lambda, \theta_2)|] \\
&\leq \mathbb{E}_{\lambda \sim \mathcal{D}}[l'(s_\lambda)\|\theta_1 - \theta_2\|] \\
&= \mathbb{E}_{\lambda \sim \mathcal{D}}[l'(s_\lambda)]\|\theta_1 - \theta_2\|,
\end{aligned}
$$

the conclusion holds and $l'_{\text{aug}} = \mathbb{E}_{\lambda \sim \mathcal{D}}[l'(s_\lambda)]$. □

**Proposition A.5.** For the pair $(\theta_{\text{aug}}, l_{\text{aug}})$, the following property holds.

$$\sup_{\theta \in \Theta} \left| \frac{1}{N+1} \sum_{i=0}^{N} l_{\text{aug}}(s_i, \theta) - \mathbb{E}_{s \sim \mathcal{P}}[l_{\text{aug}}(s, \theta)] \right| \to 0$$

*Proof.* We have

$$
\begin{aligned}
&\sup_{\theta \in \Theta} \left| \frac{1}{N+1} \sum_{i=0}^{N} l_{\text{aug}}(s_i, \theta) - \mathbb{E}_{s \sim \mathcal{P}}[l_{\text{aug}}(s, \theta)]] \right| \\
&= \sup_{\theta \in \Theta} \left| \frac{1}{N+1} \sum_{i=0}^{N} \mathbb{E}_{\lambda \sim \mathcal{D}}[l(s_{i,\lambda}, \theta)] - \mathbb{E}_{s \sim \mathcal{P}}[\mathbb{E}_{\lambda \sim \mathcal{D}}[l(s_\lambda, \theta)]] \right| \\
&= o_{\mathcal{P}}(1)
\end{aligned}
$$

where the last equality holds because the underlying sample space $\mathcal{S}$ is the disjoint countable union of all possible augmented sample spaces. □

**Proposition A.6.** For the pair $(\theta_{\text{aug}}, l_{\text{aug}})$, the following property holds. For each $\theta$ in $\Theta$, the map $s \to l_{\text{aug}}(s, \theta)$ is measurable

*Proof.* $l_{\text{aug}}$ is measurable since $l$ is measurable and $l_{\text{aug}}(s, \theta) = \mathbb{E}_{\lambda \sim \mathcal{D}}[l(s_\lambda, \theta)]$. □

**Proposition A.7.** For the pair $(\theta_{\mathrm{aug}}, l_{\mathrm{aug}})$, the following property holds. For each $\theta$ in $\Theta$, the map $s \to l_{\mathrm{aug}}(s, \theta)$ is differentiable

*Proof.* we have

$$\lim_{\delta \to 0} \frac{\left| l_{\mathrm{aug}}(s, \theta_* + \delta) - l_{\mathrm{aug}}(s, \theta_*) - \delta^T \nabla l_{\mathrm{aug}}(s, \theta_*) \right|}{\|\delta\|}$$

$$= \lim_{\delta \to 0} \frac{\left| \mathbb{E}_{\lambda \sim \mathcal{D}}[l(s_\lambda, \theta_* + \delta) - l(s_\lambda, \theta_*)] - \delta^T \mathbb{E}_{\lambda \sim \mathcal{D}}[\nabla l(s_\lambda, \theta_*)] \right|}{\|\delta\|}$$

$$= \lim_{\delta \to 0} \frac{\left| \mathbb{E}_{\lambda \sim \mathcal{D}}\left[ l(s_\lambda, \theta_* + \delta) - l(s_\lambda, \theta_*) - \delta^T (\nabla l(s_\lambda, \theta_*)) \right] \right|}{\|\delta\|}$$

$$\leq \lim_{\delta \to 0} \frac{\mathbb{E}_{\lambda \sim \mathcal{D}}\left[ \left| l(s_\lambda, \theta_* + \delta) - l(s_\lambda, \theta_*) - \delta^T (\nabla l(s_\lambda, \theta_*)) \right| \right]}{\|\delta\|}$$

Let us define

$$F_\delta(s) = \frac{\mathbb{E}_{\lambda \sim \mathcal{D}}\left[ \left| l(s_\lambda, \theta_* + \delta) - l(s_\lambda, \theta_*) - \delta^T (\nabla l(s_\lambda, \theta_*)) \right| \right]}{\|\delta\|}$$

$$G(s) = \mathbb{E}_{\lambda \sim \mathcal{D}}[|l'(s_\lambda) + (\nabla l(s_\lambda, \theta_*))|] \text{ for all } s \in S$$

Since $|F_\delta(s)| \leq G(s)$ for all $s \in S$ by Lebesgue's dominated convergence theorem,

$$\lim_{\delta \to 0} \frac{\left| l_{\mathrm{aug}}(s, \theta_* + \delta) - l_{\mathrm{aug}}(s, \theta_*) - \delta^T \nabla l_{\mathrm{aug}}(s, \theta_*) \right|}{\|\delta\|}$$

$$\leq \lim_{\delta \to 0} \frac{\mathbb{E}_{\lambda \sim \mathcal{D}}\left[ \left| l(s_\lambda, \theta_* + \delta) - l(s_\lambda, \theta_*) - \delta^T (\nabla l(s_\lambda, \theta_*)) \right| \right]}{\|\delta\|}$$

$$= \mathbb{E}_{\lambda \sim \mathcal{D}}\left[ \lim_{\delta \to 0} \frac{\left| l(s_\lambda, \theta_* + \delta) - l(s_\lambda, \theta_*) - \delta^T (\nabla l(s_\lambda, \theta_*)) \right|}{\|\delta\|} \right] = 0$$

where the last inequality holds because $l$ is differentiable. $\square$

**Proposition A.8.** The map $\theta \to \mathbb{E}_{s \sim \mathcal{P}}[l_{\mathrm{aug}}(s, \theta)]$ admits a second-order Taylor expansion at $\theta_*$ with non-singular second derivatives matrix $V_{\theta_*}$

*Proof.* By the total law of expectation of Lemma A.3, we get

$$\mathbb{E}_{s \sim \mathcal{P}}[l_{\mathrm{aug}}(s, \theta)] = \mathbb{E}_{s \sim \mathcal{P}}[l(s, \theta)].$$

Hence the conclusion holds by the bullet 6 from Assumption 3. $\square$

Combining all the Propositions [A.4 - A.8] with some results shown in Van der Vaart (1998), we prove Theorem 3.5.

***Proof of Theorem 3.5.*** The results for $\hat{\theta}$ have already been proven in (Van der Vaart, 1998, Theorem 5.23). The Propositions [A.4 - A.8] guarantee that (Van der Vaart, 1998, Theorem 5.23) can be applied to the pairs $(\theta_*, l)$ and $(\theta_{\mathrm{aug}}, l_{\mathrm{aug}})$. Therefore $\hat{\theta}_{\mathrm{aug}}$ is asymptotically normal and satisfies (Eq. 10).
Let $X = \nabla l(s, \theta_*) - \nabla l_{\mathrm{aug}}(s, \theta_*)$. Recall that $\mathrm{Cov}_{s \sim \mathcal{P}} \mu = \mathbb{E}_{s \sim \mathcal{P}}[\mathrm{Cov}(\mu | \overline{\mu})] + \mathrm{Cov}_{s \sim \mathcal{P}} \overline{\mu}$ by

**Lemma A.3.** Let's consider $\mu$ and $\overline{\mu}$ to be $\nabla l(s, \theta_*)$ and $\nabla l_{\text{aug}}(s, \theta_*)$, respectively. Then

$$\Sigma_0 = \text{Cov}(\nabla l(s, \theta_*))$$

and

$$\Sigma_{\text{aug}} = \text{Cov}(\nabla l_{\text{aug}}(s, \theta_*)).$$

Hence we have

$$
\begin{aligned}
\Sigma_0 - \Sigma_{\text{aug}} &= \text{Cov}_{s \sim \mathcal{P}} \mu - \text{Cov}_{s \sim \mathcal{P}} \overline{\mu} = \mathbb{E}_{s \sim \mathcal{P}}[\text{Cov}(\mu|\overline{\mu})] \\
&= \mathbb{E}_{s \sim \mathcal{P}}[\mathbb{E}_{s \sim \mathcal{P}}[(\mu - \mathbb{E}[\mu|\overline{\mu}])(\mu - \mathbb{E}[\mu|\overline{\mu}])^T \,|\, \overline{\mu}]] \\
&= \mathbb{E}_{s \sim \mathcal{P}}[\mathbb{E}_{s \sim \mathcal{P}}[(\mu - \overline{\mu})](\mu - \overline{\mu})^T \,|\, \overline{\mu}]] \\
&= \mathbb{E}_{s \sim \mathcal{P}}[\mathbb{E}_{s \sim \mathcal{P}}[(\nabla l(s, \theta_*) - \nabla l_{\text{aug}}(s, \theta_*))(\nabla l(s, \theta_*) - \nabla l_{\text{aug}}(s, \theta_*))^T \,|\, \overline{\mu}]] \\
&= \mathbb{E}_{s \sim \mathcal{P}} \mathbb{E}_{s \sim \mathcal{P}}[XX^T \,|\, \overline{\mu}] \\
&= \mathbb{E}_{s \sim \mathcal{P}}[XX^T].
\end{aligned}
\tag{50}
$$

Thus we get

$$\Sigma_{\text{aug}} = \Sigma_0 - \mathbb{E}_{s \sim \mathcal{P}}[XX^T]. \tag{51}$$

Since $tr(\mathbb{E}_{s \sim \mathcal{P}}[XX^T]) \geq 0$, we have $\text{RE} = \frac{tr(\Sigma_0)}{tr(\Sigma_{\text{aug}})} \geq 1$ $\qquad \square$

## B  VISUALIZATIONS OF RIM TIMEGAN GENERATED TIME SERIES

This sections shows visualization of time series generated by RIM and TimeGAN. We plot samples from original time series, RIM generated time series and TimeGAN generated time series from both classes for Synthetic exponential ODE classification in Figure 6.

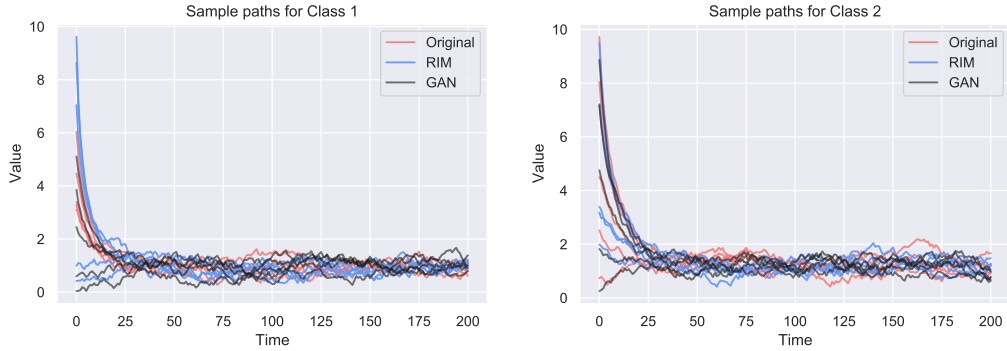

Figure 6: Visualization of exponential ODE classification series generated by RIM and TimeGAN against original series (5 each)

## C  ABLATION ANALYSIS

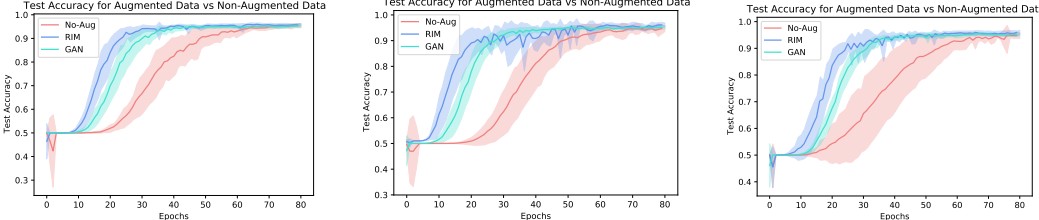

Figure 7: Test accuracy over epochs for synthetic data with Exponential ODE with $\lambda$ sampled from beta distributions with different scales (Left) $Beta(0.5, 0.5)$, (Middle) $Beta(2, 2)$, and (Right) $Beta(2, 5)$.

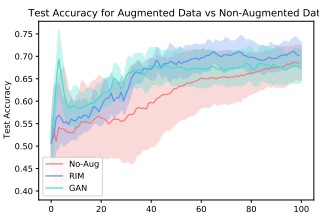 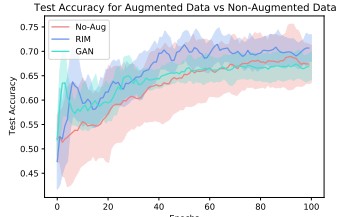 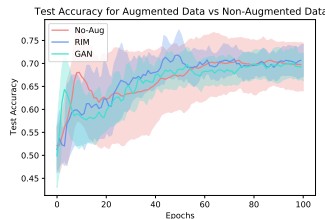

Figure 8: Test accuracy over epochs for Indoor dataset with $\lambda$ sampled from beta distributions with different scales (Left) $Beta(0.5, 0.5)$, (Middle) $Beta(2, 2)$, and (Right) $Beta(2, 5)$.

## D  TIME SERIES FORECASTING

In this section, we consider time series forecasting task where we use previous n time steps data to predict next times step data. We compare performances of regression model trained with original dataset and regression models trained with augmented dataset using RIM.

**Predicting Stock Price Movement.**
This regression task consists of predicting the next day SPY500 index Open price from historical SPY500 index using data from July 2008 to December 2012 as training data and data from January 2013 to March 2014 as testing data. The input data contains the last 7 days' historical Open, Close, High, Low, and Volume of the SPY500 index. After predicting the next day's Open price, we take a long position if the predicted next day Open is larger than today's Open, short otherwise. On comparing the results on the test data for the augmented case and the original case, we observe that the proportion of profitable trading signals is higher in the augmentation-trained model as observed in Figure 9. Using these trading signals, we also calculate the trading system's CAGR (Compound Annual Growth Rate) which we observe to be higher in the augmentation trained model. The test loss plot shows that the MSE for the augmentation trained model is consistently lower than the non-augmentation trained model.

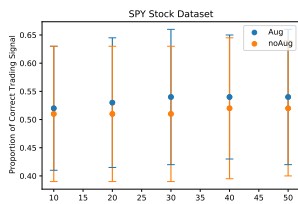 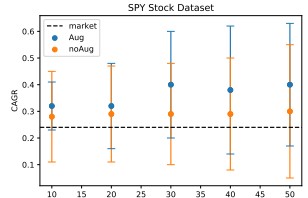 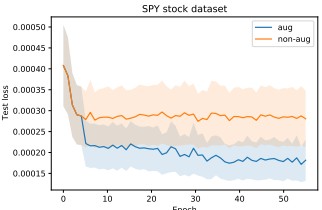

Figure 9: Profitable trading signals (left), test set CAGR (middle), test set MSE (right) for the SPY500 Dataset using an LSTM model with 2 LSTM layers (200 neurons), 2 dense layers (100 neurons), lr=1e-4, batch size=16. The plots indicate resulting mean $\pm$ standard deviation from 10 runs.

**Predicting Air Quality.**
The restricted air quality dataset contains 1200 instances of hourly averaged responses from an array of 5 metal oxide chemical sensors. This is a time series regression task where the target is the next time step's CO concentration. The input data contains the last six time steps' 10 features as used in De Vito et al. (2008) and for Machine Learning & Repository. Figure 10 shows that the test MSE of the augmentation trained model remains lower than the non-augmentation trained model during the training epochs validating our claims about the robustness of the approach. Accordingly, the proportion of correct predictions of CO up/down also remains higher for the augmented case.

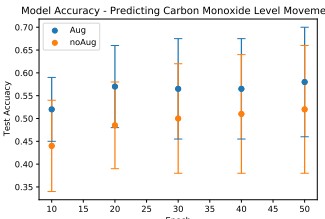 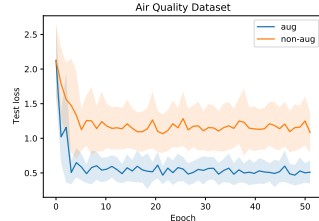

Figure 10: Test accuracy (left) and Test MSE (right) for the Air Quality Dataset using an LSTM model with 2 LSTM layers (200 neurons), 2 dense layers (100 neurons), lr=1e-4, and batch size=16. The plots indicate the resulting mean $\pm$ standard deviation from 10 runs.

# E   TIME SERIES RL

In this section, we consider portfolio management task using reinforcement learning. More specifically, we compare performances of agents (DPG) trained with original state trajectories (price evolution) and augmented state trajectories using RIM.

## E.1   DATASET

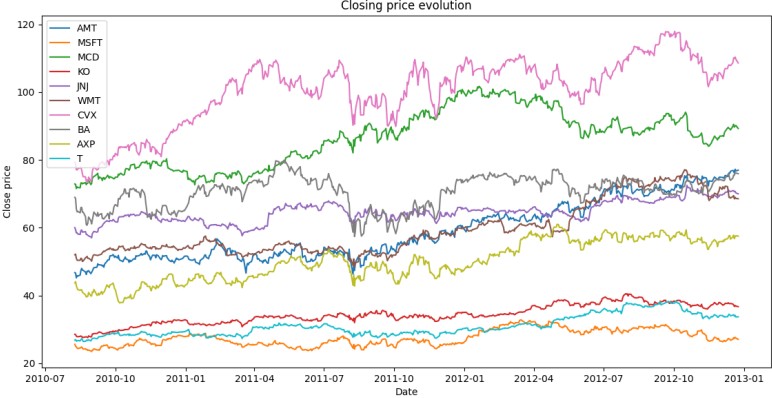

Figure 11: Evolution of stock prices.

For RL, the data comes from the Quandl finance database that has daily data. Our RL models are trained over 600 trading days from 2010-08-09 to 2012-12-25 and tested over 200 trading days from 2012-12-26 to 2013-10-11 on a portfolio consisting of ten selected stocks: American Tower Corp. (AMT), American Express Company (AXP), Boeing Company (BA), Chevron Corporation (CVX), Johnson & Johnson (JNJ), Coca-Cola Co (KO), McDonald's Corp. (MCD), Microsoft Corporation (MSFT), AT&T Inc. (T) and Walmart Inc (WMT). To promote the diversification of the portfolio, these stocks were selected from different sectors of the S&P 500, so that they are uncorrelated as much as possible as shown in Fig. 11.

## E.2 RL PSEUDOCODE

---

**Algorithm 1:** RIM: RL Training

---

**input** : Observed time series data $S = \{s_0, s_1, \ldots, s_T\}$ where $s_i = (x_i, y_i)$ for $x_i \in \mathbb{R}^d$ and $y_i \in \mathbb{R}$ for $i \in [0 : T]$.

1 **Initialize** $\theta$ parameter for the policy network, $Y$ epochs, and a distribution $\mathcal{D}$ with support $[0, 1)$.

**for** $e = 1$ *to* $Y$ **do**

2      **Initialize** $\vec{\lambda} = (\lambda_1, \ldots, \lambda_T)$ with $\lambda_i \sim \mathcal{D}$ // Initialize interpolation coefficients vector

     **Augmented Path Simulator** Generate an augmented trajectory $S_{\vec{\lambda}} = \{s_0, s_{1,\lambda_1}, \ldots, s_{T,\lambda_T}\}$ where $s_{i,\lambda_i} = (x_{i,\lambda_i}, y_{i,\lambda_i})$

     **for** $t = 1$ *to* $T$ **do**

3          $a_t \sim \pi(.|s_{t,\lambda_t})$

4          $s'_t \sim p(.|s_{t,\lambda_t}, a_t)$

5          $S_t \leftarrow S \cup S_{\vec{\lambda}}$ // Add a transition to the replay buffer

6          UpdateCritic($S_t$)

7          UpdateActor($S_t$) // Data augmentation is applied to the samples for actor training as well

8      **end**

9 **end**

---

## E.3 POLICY DEPLOYMENT

In our RL experiments, the investment decisions to rebalance the portfolio are made daily and each input signal represents a multidimensional tensor that aggregates historical open, low, high, close prices and volume. It should be noted that our training and testing include the transaction costs (TC). We used the typical cost due to bid-ask spread and market impact that is 0.25%. We believe these are reasonable transaction costs for the portfolio trades.

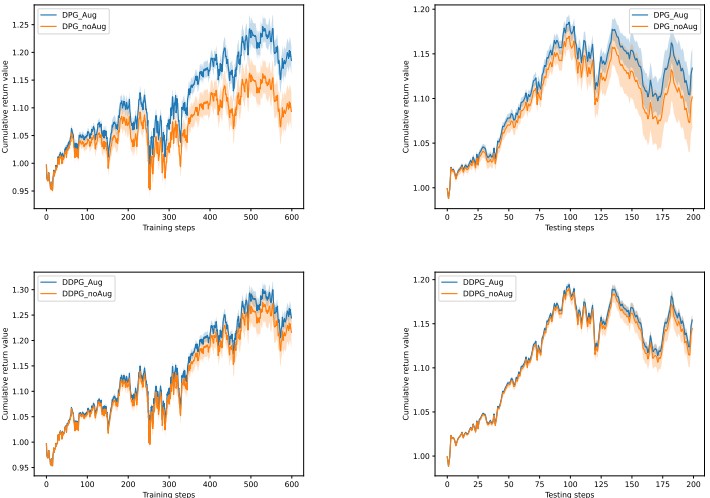

Figure 12: Training and testing results for DPG (above) and DDPG (below). The plots indicate the resulting mean ± standard deviation from 20 runs with different seeds.

# F  HYPERPARAMETERS

We note that the primary objective of the conducted experiments is to show that RIM can improve model performance. Therefore, in all experiments, instead of finding optimal set of parameters for augmented and non-augmented trained model, we compare the performance of augmented trained model and non-augmented trained model with the same hyperparameter configuration. We demonstrated in Section 4 that RIM indeed improves model performance with same hyperparameter configuration. However, are these improvements robust to other hyperparameters? To answer this question, we conducted sensitivity analysis for two supervised learning tasks (Indoor user movement classification and Air Quality regression) and RL task (Portfolio Management).

## F.1  HYPERPARAMETER SENSITIVITY FOR SUPERVISED TASKS

For Indoor movement classification task, we conducted the same experiment in Section 4 with 9 different hyperparameter configurations as shown in table 1 and observed that for all the cases RIM outperforms Non-Augmented case (with smaller mean test loss and higher mean test accuracy) which solidifies our claim of enhancement observed in model performance when we use RIM.

Table 1: In the table, the Test Loss and Test Accuracy are the mean(standard deviation) over 10 runs for 50 epochs for varying Filters and Kernel size for Indoor User Movement Classification Task

| Filters | Kernel Size | Test Loss – RIM | Test Loss – NoAug | Test Acc – RIM | Test Acc – NoAug |
|---|---|---|---|---|---|
| 16 | 3 | **1.09 (0.43)** | 2.72 (0.95) | **0.71 (0.04)** | 0.63 (0.02) |
| 16 | 4 | **0.90 (0.45)** | 1.70 (0.69) | **0.76 (0.04)** | 0.68 (0.04) |
| 16 | 5 | **0.86 (0.19)** | 1.00 (0.19) | **0.72 (0.01)** | 0.60 (0.05) |
| 32 | 3 | **1.37 (0.31)** | 3.07 (0.67) | **0.72 (0.02)** | 0.62 (0.05) |
| 32 | 4 | **1.75 (0.71)** | 2.70 (1.20) | **0.74 (0.03)** | 0.68 (0.03) |
| 32 | 5 | **1.70 (0.70)** | 2.57 (0.85) | **0.66 (0.08)** | 0.64 (0.06) |
| 64 | 3 | **2.99 (2.17)** | 3.31 (1.36) | **0.68 (0.07)** | 0.68 (0.06) |
| 64 | 4 | **2.60 (0.70)** | 4.80 (4.30) | **0.73 (0.03)** | 0.68 (0.02) |
| 64 | 5 | **3.30 (0.74)** | 4.90 (4.08) | **0.66 (0.03)** | 0.60 (0.06) |

For Air quality regression task, we again conducted the same experiment as in Section 4 with 8 different hyperparameter configurations as shown in table 2. Here too, we find that RIM outperforms Non-Augmented case (with smaller mean test MSE and higher mean test accuracy) which confirms our claim of enhancement observed in model performance when we use RIM.

Table 2: In the table, the MSE and Accuracy are the mean(standard deviation) over 10 runs for 50 epochs for varying Epoch Initial, Batch Size, LSTM Layer and Dense Layer for Air Quality Regression Task on Test Data

| Epoch Init | Batch Size | LSTM Layer | Dense Layer | MSE – RIM | MSE – NoAug | Acc – RIM | Acc – NoAug |
|---|---|---|---|---|---|---|---|
| 5 | 16 | 100 | 50 | **5.34 (0.11)** | 5.43 (0.25) | **0.63 (0.03)** | 0.47 (0.08) |
| 1 | 32 | 80 | 40 | **5.40 (0.08)** | 5.44 (0.04) | **0.63 (0.05)** | 0.51 ( 0.02) |
| 5 | 16 | 150 | 50 | **5.59 (0.01)** | 5.74 (0.35) | **0.66 (0.03)** | 0.53 (0.10) |
| 6 | 32 | 200 | 100 | **5.23 (0.10)** | 5.55 (0.19) | **0.64 (0.02)** | 0.51 (0.06) |
| 10 | 16 | 150 | 100 | **5.32 (0.05)** | 5.48 (0.11) | **0.57 (0.02)** | 0.51 (0.35) |
| 1 | 64 | 200 | 100 | **5.59 (0.07)** | 5.67 (0.23) | **0.65 (0.02)** | 0.52 (0.03) |
| 10 | 16 | 100 | 50 | **5.44 (0.24)** | 5.49 (0.38) | **0.59 (0.05)** | 0.56 (0.07) |
| 15 | 16 | 100 | 50 | **5.56 (0.08)** | 5.60 (0.16) | **0.65 (0.01)** | 0.56 (0.02) |

## F.2 Hyperparameter Sensitivity for RL Task

The hyperparameter space is represented by a hypercube: the more values it contains the harder it is to explore all the possible combinations. To efficiently find the optimal set of hperparameters, we explored the hyperparameter space using Bayesian optimization (BO) Hutter et al. (2011). Table 3 shows the range of values for the hyperparameters used during the training and validation phase. The learning rate controls the speed at which neural network parameters are updated. The window is used to allow the deep RL agents to utilize a range of historical data values to relax the Markov assumption. We allow the use of 2 days up to 30 days of historical data. The number of filters and kernel strides are the hyperparameters for the convolution neural networks. It is important to carefully optimize these parameters in order to capture the best feature representations used by the policy networks. Finally, the training and testing sizes may also impact the RL performance. So, we also consider them as hyperparameters.

Table 3: Hyperparameters used by our RL algorithms.

| Parameters | Bounds | Type |
|---|---|---|
| Learning rate (lr) | $10^{-5}$–$5.10^{-1}$ | Discrete |
| Trading cost (tc) | 2–30 | Discrete |
| Number of filters (nf) | 2–52 | Discrete |
| Kernel Strides (ks) | 2–10 | Discrete |
| Window | 2–30 | Discrete |
| Training size (train) | 20–500 | Discrete |
| Testing size (test) | 5–100 | Discrete |

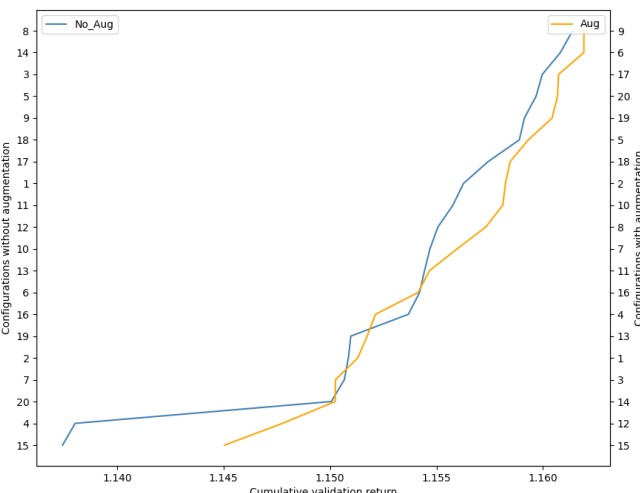

Figure 13: Hyperparameter sensitivity for DPG: the vertical axes list all the models we evaluated by its index. The detailed hyperparameter configurations each index refers to are listed in Tables 4 and 5. The horizontal axis shows the cumulative total validation return. The blue line shows the validation performance for DPG without augmentation. The orange line shows the validation performance for DPG using RIM. The worst to best models are ordered from bottom to top.

Table 4: Sensitivity analysis for DPG configurations without augmentation

| lr | tc | nf | ks | window | train | test | validation cumulative return |
|---|---|---|---|---|---|---|---|
| 0.001 | 0.0225 | [48, 20, 1710] | 9 | 19 | 50 | 10 | 1.1563 |
| 1.0 | 0.0125 | [50, 38, 1340] | 3 | 6 | 500 | 100 | 1.1509 |
| 0.0005 | 0.0125 | [14, 6, 1300] | 4 | 16 | 50 | 50 | 1.1600 |
| 0.01 | 0.0125 | [28, 28, 210] | 4 | 13 | 100 | 10 | 1.1380 |
| 0.01 | 0.0175 | [16, 2, 1080] | 3 | 9 | 50 | 5 | 1.1597 |
| 0.001 | 0.0125 | [18, 12, 1290] | 8 | 16 | 50 | 50 | 1.1542 |
| 1.0 | 0.0175 | [20, 20, 1090] | 6 | 16 | 50 | 10 | 1.1507 |
| **0.0005** | **0.0075** | **[18, 2, 1270]** | **3** | **16** | **50** | **50** | **1.1614** |
| 1,00E-05 | 0.0225 | [10, 10, 1310] | 8 | 9 | 50 | 50 | 1.1591 |
| 0.1 | 0.0225 | [10, 10, 770] | 2 | 17 | 200 | 20 | 1.15470 |
| 5,00E-05 | 0.0075 | [12, 16, 1300] | 7 | 17 | 50 | 50 | 1.1558 |
| 0.05 | 0.0125 | [16, 10, 1090] | 3 | 7 | 50 | 5 | 1.1551 |
| 0.0001 | 0.0075 | [24, 12, 1080] | 5 | 15 | 50 | 5 | 1.1544 |
| 0.1 | 0.0175 | [44, 2, 1500] | 2 | 11 | 20 | 10 | 1.16082 |
| 0.005 | 0.0025 | [2, 42, 270] | 8 | 16 | 500 | 50 | 1.1374 |
| 5,00E-05 | 0.0175 | [24, 10, 1090] | 2 | 11 | 50 | 5 | 1.1537 |
| 1.0 | 0.0175 | [32, 12, 1490] | 2 | 8 | 20 | 20 | 1.1574 |
| 0.5 | 0.0075 | [46, 6, 950] | 4 | 18 | 20 | 20 | 1.1589 |
| 0.0001 | 0.0025 | [30, 26, 1190] | 7 | 9 | 500 | 100 | 1.1510 |
| 0.01 | 0.0025 | [50, 16, 940] | 7 | 15 | 20 | 5 | 1.1501 |

Table 5: Sensitivity analysis for DPG configurations with augmentation

| lr | tc | nf | ks | window | train | test | validation cumulative return |
|---|---|---|---|---|---|---|---|
| 0.005 | 0.0175 | [36, 36, 1960] | 4 | 9 | 20 | 10 | 1.1513 |
| 0.0001 | 0.0025 | [22, 14, 1270] | 9 | 10 | 50 | 50 | 1.1582 |
| 0.1 | 0.0125 | [40, 22, 880] | 3 | 6 | 20 | 10 | 1.1503 |
| 1,00E-05 | 0.0075 | [8, 16, 760] | 2 | 6 | 200 | 20 | 1.1521 |
| 0.005 | 0.0175 | [44, 6, 1500] | 5 | 6 | 20 | 20 | 1.1593 |
| **1,00E-05** | **0.0025** | **[2, 2, 1310]** | **2** | **19** | **50** | **50** | **1.1619** |
| 0.001 | 0.0125 | [36, 6, 960] | 3 | 17 | 20 | 20 | 1.1560 |
| 0.0001 | 0.0075 | [22, 6, 770] | 9 | 13 | 200 | 20 | 1.1573 |
| **1,00E-05** | **0.0025** | **[22, 2, 1320]** | **2** | **19** | **50** | **50** | **1.1619** |
| 0.5 | 0.0225 | [28, 2, 970] | 7 | 12 | 20 | 5 | 1.1581 |
| 0.5 | 0.0125 | [40, 14, 960] | 9 | 18 | 20 | 10 | 1.1547 |
| 0.01 | 0.0175 | [32, 16, 760] | 5 | 7 | 200 | 20 | 1.1478 |
| 0.01 | 0.0025 | [24, 10, 780] | 7 | 16 | 200 | 5 | 1.1517 |
| 0.005 | 0.0025 | [46, 12, 770] | 9 | 19 | 200 | 20 | 1.1502 |
| 0.01 | 0.0225 | [48, 44, 1190] | 4 | 10 | 100 | 50 | 1.1450 |
| 1.0 | 0.0175 | [28, 8, 960] | 6 | 13 | 20 | 5 | 1.1541 |
| 0.005 | 0.0125 | [36, 2, 1500] | 3 | 15 | 20 | 5 | 1.1607 |
| 0.005 | 0.0225 | [18, 8, 1270] | 5 | 9 | 50 | 50 | 1.1585 |
| 1.0 | 0.0125 | [8, 6, 1260] | 7 | 8 | 50 | 50 | 1.1604 |
| 1.0 | 0.0025 | [28, 2, 1250] | 2 | 7 | 50 | 50 | 1.1607 |

