# OpenReview forum: "Recursive Time Series Data Augmentation"
_ICLR.cc/2023/Conference — ICLR 2023 poster_

### Official Review · Reviewer_Ftkr · 2022-10-24

**Confidence:** 3
**Clarity, Quality, Novelty And Reproducibility:** Good
**Correctness:** 3
**Technical Novelty And Significance:** 3
**Empirical Novelty And Significance:** 3
**Recommendation:** 6

**Strength And Weaknesses:**

Pros:

(1) The motivation to perform data augmentation that preserves the original inherent dynamics makes sense.

(2) The methodology is clearly described and theoretical analysis is given.

(3) Experiments are extensively conducted and it is shown that the model with data augmentation clearly outperforms the model without augmentation.

Cons:

(1) Same to my concerns in the previous review, the method in this paper assume that the data is stationary, which is not the case in real world. Though it is possible to use a large lambda close to 1 for non-stationary data, I still feel the assumption made in the method is a drawback.

**Summary Of The Paper:**

I have reviewed this paper in another conference.
This paper introduces a technique called Recursive Interpolation Method (RIM) for time series augmentation. The augmented samples preserve the original inherent time-series dynamics. Theoretical analysis is given to guarantee the test performance. Experiments are conducted on multiple time series tasks such as classification, regression and RL.

**Summary Of The Review:**

A good paper with theoretical insights and experimental support. But a flaw in the assumption of the model.

---

> ### Author Response · Authors · 2022-11-19
> **Comment for Reviewer Ftkr**
>
> Same to my concerns in the previous review, the method in this paper assume that the data is stationary, which is not the case in real world. Though it is possible to use a large lambda close to 1 for non-stationary data, I still feel the assumption made in the method is a drawback.
>
> - In our framework, we do not assume that the data is stationary. Whenever we are given time series data, we augment the data along the time axis. Empirically, we experimented with non-stationary data like stock market data and demonstrated that our method achieves a better performance than the non augmented case. When we know our data is non stationary, we can sample $\lambda$ from distributions skewed towards higher values closer to 1, which results in wider bounds in equations 4, 5, and 6 but still gives good and useful results. Extending our methodology to the more general theoretical non-stationary setting is definitely an interesting topic. We plan to pursue this research avenue in our future work. Thank you very much for your constructive comment.

---

> ### Comment · Reviewer_Ftkr · 2022-12-01
> **Thank you for your response**
>
> I've read the response and other reviewers' comments. Given that the proposed approach can improve performance even on non-stationary datasets, I'd like to keep my score unchanged.

---

### Official Review · Reviewer_oRTw · 2022-10-24

**Confidence:** 3
**Correctness:** 3
**Technical Novelty And Significance:** 3
**Empirical Novelty And Significance:** 3
**Recommendation:** 5

**Clarity, Quality, Novelty And Reproducibility:**

- The paper tackles an interesting problem, and I appreciated that the authors went beyond simply empirically verifying the benefit of the approach by also presenting theoretical guarantees on the expected quality of the augmented time series.
- A highlighted earlier, it would benefit from more illustrative examples and figures to properly convey which properties of time series influence the quality of the recursive procedure.
- While extensive, the experiment section could also be improved by adding more ablation studies and more insightful discussions on the results.
- The paper contains a few minor typos, but these should be easy to identify and fix.


**Strength And Weaknesses:**

- The recursive approach detailed in this work is intuitive, and I appreciated how it presents a fairly straightforward solution to a problem that manifests itself in any domain where there is risk of overfitting to limited representative time series.
- The paper is mostly well-written, although there were several instances where I wished for more clarity or detail. I especially think that the authors should consider including more illustrative examples within the main paper itself such that the concepts become easier to follow. It is otherwise difficult to sometimes follow which time series structure the authors are referring to throughout the text, and it could also better illustrate the circumstances under which the guarantees put forward in the paper would not hold.
- One of the main motivations for undertaking this work was that existing augmentation techniques are problem dependent, so I would have liked to see more concrete examples of this. I consider such insight and analysis to be a crucial contribution of the paper, and yet it currently feels very underdeveloped.
- Another appealing aspect of this work is that the augmented time series can be leveraged in multiple different downstream tasks, which are highlighted in the introduction to the paper. However, besides a small concluding interlude in Section 4.3, experiments on next step prediction and reinforcement learning are left entirely to the Supplementary Material, with the focus in the main paper being exclusively on classification tasks. While there is a nice variety of synthetic and real-world classification problems here, featuring a different task entirely would have given a more complete picture.
- The actual discussion of the results is also quite surface-level, and I would have liked to see more insights provided throughout as well as ablation studies that tweak different aspects of the augmentation procedure.

**Summary Of The Paper:**

Training set augmentation has been found to be an effective way of improving performance on downstream tasks such as classification accuracy, particularly for computer vision problems. Augmentation has also been considered for the time series domain, although much of the contributions in this space implicitly make assumptions on the structure of the time series, and can thus be less widely applied. It is also difficult to provide guarantees on the quality of generated samples using conventional methods. In this work, the authors propose a recursive approach for generating time series that follow the same dynamic process as a (possibly) limited number of observed time series, for which it is also possible to obtain guarantees on quality under certain conditions. The authors demonstrate how the approach compares favorably to augmentation methods leveraging GANs, while also being more computationally efficient.

**Summary Of The Review:**

The contributions featured in this paper are interesting, and provide an intuitive way of overcoming sparsity in time series datasets. However, I believe there is still considerable room for improvement in the paper, and I encourage the authors to consider rewriting certain sections in order to include further illustrative example and ablation studies that will make the paper more complete and insightful to both readers and practitioners.


** Post-rebuttal comments **

Thank you for your replies, and the updates to the paper. I am raising my score to a '5', although I still think this paper would benefit from another round of reviewing. The ablation study included as part of the rebuttal is interesting, but I believe there could still be more clarity on aspects such as settings where the RIC approach doesn’t perform as well.

---

> ### Author Response · Authors · 2022-11-19
> **Comment for Reviewer oRTw - Q1**
>
> The recursive approach detailed in this work is intuitive, and I appreciated how it presents a fairly straightforward solution to a problem that manifests itself in any domain where there is risk of overfitting to limited representative time series.
>
> - Thank you very much for your comment.

---

> ### Author Response · Authors · 2022-11-19
> **Comment for Reviewer oRTw - Q2**
>
> The paper is mostly well-written, although there were several instances where I wished for more clarity or detail. I especially think that the authors should consider including more illustrative examples within the main paper itself such that the concepts become easier to follow. It is otherwise difficult to sometimes follow which time series structure the authors are referring to throughout the text, and it could also better illustrate the circumstances under which the guarantees put forward in the paper would not hold.
>
> - For illustrative purposes, we have added a figure (Figure 1 on page 3 of the main paper) depicting how our approach works. Based on Theorem 3.1, we guarantee that the augmented time series sample lies within some distance bound of the original time series sample. In addition, Theorem 3.1 has some key results closely connected to Theorems 3.2, 3.3 and 3.4, which involve the theoretical results for learning parameters.

---

> ### Author Response · Authors · 2022-11-19
> **Comment for Reviewer oRTw - Q3**
>
> One of the main motivations for undertaking this work was that existing augmentation techniques are problem dependent, so I would have liked to see more concrete examples of this. I consider such insight and analysis to be a crucial contribution of the paper, and yet it currently feels very underdeveloped.
>
> - Thank you for the comment. However, we are not sure if we understand the question. If you can further clarify your concern, we will be very happy to take it into consideration.

---

> > ### Comment · Reviewer_oRTw · 2022-11-23
> > **Clarification**
> >
> > Thank you for your replies to the rebuttal. My point here was mostly focused on the claim towards the end of page 2 that _"These approaches are problem dependent and do no offer theoretical guaranteed learning improvement"_.
> >
> > This statement wasn't altogether clear to me, and I think the paper could possibly include additional examples that indicate this constraint more clearly. I suppose my question also centres on what exactly is mean by "problem dependent" in this context.

---

> > > ### Author Response · Authors · 2022-12-12
> > > **Response to Clarification**
> > >
> > > Thank you for your clarification. The augmentation approaches mentioned in Section 2 (Related Work) have been developed for specific problems and do not necessarily make sense for other problems. For example, rotation makes sense for some problem domains as highlighted in 'Time Series Data Augmentation for Neural Networks by Time Warping with a Discriminative Teacher' by Brian Kenji Iwana and Seiichi Uchida in 2020. However, a rotated time series (flipped) does not make sense for financial time series because a rotated financial time series does not have the same meaning. The key aspect of our time series augmentation RIM is that it is problem independent because we can control the deviation of the augmentation from the original time series with the lambda parameter to ensure that the original label is still valid (please refer to the newly added Figure 1). Furthermore, other time series augmentations do not have theoretical guarantee for faster learning convergence with reduced variance as provided in our time series augmentation methodology RIM. Our theoretical framework allows us to test a dataset to determine if the deviation is too large in which case, we can't guarantee the sample efficiency and better convergence but we can guarantee that it will not be worse than the non augmented results as shown in Appendix C.

---

> ### Author Response · Authors · 2022-11-19
> **Comment for Reviewer oRTw - Q4**
>
> Another appealing aspect of this work is that the augmented time series can be leveraged in multiple different downstream tasks, which are highlighted in the introduction to the paper. However, besides a small concluding interlude in Section 4.3, experiments on next step prediction and reinforcement learning are left entirely to the Supplementary Material, with the focus in the main paper being exclusively on classification tasks. While there is a nice variety of synthetic and real-world classification problems here, featuring a different task entirely would have given a more complete picture.
>
> - We decided to keep all time series classifications experiments in the main text and other experiments on the appendix due to space constraints and to improve clarity. Following your suggestion, we could move some classification tasks into Appendix and move reinforcement learning and regression tasks into the main text; or alternatively we could put all plots of experiments into Appendix and only summarize all results in a table in the main text, thus all tasks can be included in the main text. Do you have a recommendation?

---

> ### Author Response · Authors · 2022-11-19
> **Comment for Reviewer oRTw - Q5**
>
> The actual discussion of the results is also quite surface-level, and I would have liked to see more insights provided throughout as well as ablation studies that tweak different aspects of the augmentation procedure.
>
> - Following your comment, we ran more ablation experiments and added the results in Appendix C. Since $\lambda$ is the only parameter used in our RIM augmentation technique, our ablation study paid very precise attention to the choice of the $\lambda$ parameter. Under this, we tested different $\lambda$ distributions. Given that we were interested in convex combinations of two observations, we had to restrict $\lambda$ between 0 and 1. Two ways to perform this would be: (1) uniformly distribute the weights while sampling $\lambda$; (2) concentrating on a specific part of $\lambda$ distribution. To address (1), we use $U(0,1)$ which is the main test-bed for all the experiments in the current main text. Whereas to address the (2), we here perform studies using beta distribution by varying its shape parameters to focus on specific parts of densities. We tested beta(2,2), beta(0.5,0.5) and beta(2,5), for which the resulting plots are now added in Appendix C. Here too, we observed improvements from using RIM compared to non-augmented training, both in terms of a higher final testing accuracy and with fewer training iterations. Besides this, we also conducted hyperparameter sensitivity analysis in Appendix F.

---

### Official Review · Reviewer_aEoM · 2022-10-24

**Confidence:** 3
**Clarity, Quality, Novelty And Reproducibility:** see above
**Correctness:** 3
**Technical Novelty And Significance:** 3
**Empirical Novelty And Significance:** 2
**Recommendation:** 5

**Strength And Weaknesses:**

Pros:
1. Data augmentation is critical for time series analysis. We have seen a lot of research for developing new data augmentation methods to improve the performance of downstream tasks, but few of them provided theoretical analysis. This work opens the door for analyzing the generalization capability of time series data augmentation. Besides, this work also presents a new data augmentation method based on recursive interpolation.

2. Empirical studies verify the efficacy of the proposed data augmentation method on both synthetic and real-world datasets.


Cons:

Major:
1. The proposed RIM in sec 3.1 lacks motivation. It would be better to provide more motivational examples showing how it can benefit the time series related tasks. Most of the existing data augmentation methods aim to enhance the temporal dependencies, but the proposed RIM in Eq 1. is temporal independent. I am unclear on how  RIM can help avoid overfitting and capture better temporal patterns. Since the proposed data augmentation method is temporal independent, it can be generalized to any tabular data. How does this method connect to time series data? Additionally, RIM in eq. 1 is a kind of mixup of two consecutive dimensions. How to justify its correctness, especially when there exists dependency over two non-consecutive dimensions?

2. The theoretical analysis is interesting but needs more clarification. It is clear to me the effect of sample size and the distance between the original time series and the augmented one. Still, it is unclear how to control and decrease the Rademacher complexity for the augmented data. Eq.6 does not guide how the proposed data augmentation method can reduce it.

3. The empirical studies are not promising. First, I am not sure why the authors do not compare with other data augmentation methods, but the time series generative model, TimeGAN. Besides, there are a lot of novel generative time series models, like the generative Fourier model,  the diffusion model beat TimeGAN with a margin. Finally, since the major contribution of this work is about the generalization capability, it would be more interesting to present the empirical analysis for each complexity term to show its vacuousness.

4. The proposed method is limited to point-wise time series classification, and it can not generalize to the more common settings, like window-wise classification or conventional time series forecasting problems. This might reduce the impact of this work.

Minor:
1. what's the definition of 'dem'?
2. It would be better to present assumptions 3 and 4 in the main content to make the whole manuscript self-contained.

**Summary Of The Paper:**

This work propose a new time series augmentation technique, called Recursive Interpolation Method (RIM), to learn accurate models with limited data. They provide theoretical analysis with the guaranteeing faster convergence with reduced variance. Empirically, they show that the proposed method improves the performance on classification tasks over the SOTA on both real-world and synthetic datasets.

**Summary Of The Review:**

This paper studies an interesting and important problems, but the proposed method lacks motivation

---

> ### Author Response · Authors · 2022-11-19
> **Comment for Reviewer aEoM - Q1**
>
> The proposed RIM in sec 3.1 lacks motivation. It would be better to provide more motivational examples showing how it can benefit the time series related tasks. Most of the existing data augmentation methods aim to enhance the temporal dependencies, but the proposed RIM in Eq 1. is temporal independent. I am unclear on how RIM can help avoid overfitting and capture better temporal patterns. Since the proposed data augmentation method is temporal independent, it can be generalized to any tabular data. How does this method connect to time series data? Additionally, RIM in eq. 1 is a kind of mixup of two consecutive dimensions. How to justify its correctness, especially when there exists dependency over two non-consecutive dimensions?
>
> - The primary motivation for our work stems from the scarcity of data, which is often a problem in the time-series domain. Under such circumstances, we often resort to severe over-fitting thereby preventing generalization. To address this issue, we introduce this framework for time series augmentation and perform theoretical analysis to characterize the proposed framework and to guarantee its performance under certain conditions. An evident use-case can be a stock's time series where we only have one realization and we would like to learn the underlying dynamics. For this, any augmented data within a bound from the original data representing the original evolution can help. And that is indeed the primary motivation for our framework; to augment the original data for generalizing the learning by ensuring that the augmented data is within a specific distance bound of the original data. In the appendix, we do indeed experiment with stock market data using reinforcement learning and regression (two downstream tasks) and also show ablation studies (Table 4 and Table 5). For such use-cases where we have limited data [i.e., performance of machinery with limited inventory or defaults within a new machine over time], or single realization [i.e., single time series realization for stock price (predicting next window's stock price), weather evolution (predicting next window's weather)], people often use overlapping/sliding  windows of such time series to augment the number of training samples. However, given that they can be of different regimes under different windows (i.e., a specific stock might not behave similarly in 2010 and 2022), that approach breaks down. This is where an approach preserving the continuous dynamics helps which is what RIM provides.
>
> - The proposed RIM in Eq.1 is time dependent, where $i$ is the time dimension. The term $x_{i-1, \lambda_{i-1}}$ captures temporal history from all previous time increments. We modified the paper to make it clear that $i$ represents time.
>
> - We avoid overfitting by using the parameter $\lambda$ in Eq.4 to ensure that the augmentations are not too near nor too far from the original time series. Our RIM method can be generalized to tabular data using different interpretations of the equations. For time series data, we can guarantee that the accumulated features along time axis have some information on the time series data to represent the trend inherent in the data. Now, if we use such a method on tabular data, then the distance between the original observation and the augmented one will become a similarity measure between the distribution of the features, and the techniques and interpretation involved in the theoretical framework will be different which is beyond the scope of the current paper.
>
> - The recursion property used in RIM accumulates past information through time. In other words, our method can capture dependency between non-consecutive time series dimensions. So, for a time series with $n$ dimensions, observation $k$, where $k < n$, depends on observation $k-1$ from equation 1, which then again depends on dimension $k-2$ and so on. As a result, every observation will have some weight from the previous observations which eventually helps us retain the dependency across time for non-consecutive observations.

---

> ### Author Response · Authors · 2022-11-19
> **Comment for Reviewer aEoM - Q2**
>
> The theoretical analysis is interesting but needs more clarification. It is clear to me the effect of sample size and the distance between the original time series and the augmented one. Still, it is unclear how to control and decrease the Rademacher complexity for the augmented data. Eq.6 does not guide how the proposed data augmentation method can reduce it.
>
> - Following your comment, we made certain changes on how we present our Theorem 3.2. Under this redefined version, we show how the difference in the Rademacher complexities (before and after augmentation) is upper bounded solely by the characteristics of the time series. Equation 6 expresses how the Rademacher complexity of the augmented loss is related to the Rademacher complexity of the non augmented loss with an additional term that captures the deviation between the two time series. When the deviation term is small (the two samples are close to each other), the Rademacher of the augmented loss is always smaller than Rademacher of the non augmented loss which means that our augmented learning method (RIM) has a better learning convergence. On the contrary, when this deviation term is large, we can not guarantee that training with augmented loss will lead to a better learning convergence. Therefore, in order to guarantee that the training using augmentation has a better convergence, we can better control this deviation term by controlling the $\lambda$ (which is used to augment data using RIM). This is the main motivation behind Theorem 3.1 and this is how Theorem 3.1 connects to Theorem 3.2.

---

> ### Author Response · Authors · 2022-11-19
> **Comment for Reviewer aEoM - Q3**
>
> The empirical studies are not promising. First, I am not sure why the authors do not compare with other data augmentation methods, but the time series generative model, TimeGAN. Besides, there are a lot of novel generative time series models, like the generative Fourier model, the diffusion model beat TimeGAN with a margin. Finally, since the major contribution of this work is about the generalization capability, it would be more interesting to present the empirical analysis for each complexity term to show its vacuousness.
>
> - Generative models (diffusion model, score-based model, all GAN techniques) require relatively large amount of data from original distribution p(x), even with noise perturbation, to be able to learn an accurate reverse diffusion probability transition function in diffusion model, or the score function in score-based model. As a result, a limited data regime would lead to poor time series generation. For this reason, i.e., limited data in our experiments, timeGAN leads to poor downstream task performance. For experimentation purposes, we have four classification tasks with only a small amount of data, one realization (observation) for the regression task (air quality history, stock market history), and one realization (observation) for the reinforcement learning task (stock price history). Under all these scenarios, the above mentioned methods can not work (diffusion model, score-based model and GAN techniques), but RIM can work as it can generate multiple time series from just one series by our controlled recursive process with different sampled $\lambda$'s.
>
> - Thank you for the comment. Appendix F presents an empirical analysis of the optimization of the hyperparameters that control the complexity of the machine learning models. Tables 4 and 5 show the relationship between the neural network size and performance.

---

> ### Author Response · Authors · 2022-11-19
> **Comment for Reviewer aEoM - Q4**
>
> The proposed method is limited to point-wise time series classification, and it can not generalize to the more common settings, like window-wise classification or conventional time series forecasting problems. This might reduce the impact of this work.
>
> - We have indeed considered these settings. However, given the space constraints, we have some of them in the Appendix. Specifically, in Appendix D and E, we tackle regression and RL tasks. Within regression, we have two cases, where we have window-wise forecast (stock market) and point-wise forecast (air quality). For RL, we have window-wise learning (stock market).

---

> ### Author Response · Authors · 2022-11-19
> **Comment for Reviewer aEoM - Minor comments**
>
> What's the definition of 'dem'?
> It would be better to present assumptions 3 and 4 in the main content to make the whole manuscript self-contained.
>
> - Thank you for pointing this out. We addressed your comments in the new attached version of our draft paper.

---

### Official Review · Reviewer_LCdk · 2022-10-25

**Confidence:** 4
**Clarity, Quality, Novelty And Reproducibility:** Very clear and solid work.
**Correctness:** 4
**Technical Novelty And Significance:** 4
**Empirical Novelty And Significance:** 4
**Recommendation:** 10

**Strength And Weaknesses:**

Strengths:
1. The pipeline is based on a principled approach.
2. Stochastic type loss functions are used (cross-entropy) instead of deterministic ones.


**Summary Of The Paper:**

The authors present a time-series augmentation method names Recursive Interpolation Method (RIM), designed to boost performance in learning tasks with limited data. The method is solidly based on a theoretical based, and the analytic results are constructed with numerical experiments that prove the goodness of the technique.

**Summary Of The Review:**

The authors present a time-series data augmentation approach based on theoretical results. The work is solid and checked against experiments. This kind of work is in stark contrast to most of what can be seen in the field: ad-hoc pipelines with some experimental testing where it is almost impossible to assess the generality of the results.

---

> ### Author Response · Authors · 2022-11-19
> **Comment for Reviewer LCdk**
>
> Thank you for your kind comments.

---

### Decision · Program_Chairs · 2023-01-20

**Decision:**

Accept: poster

**Justification For Why Not Higher Score:**

It is far below the threshold for spotlight acceptance unless the major weaknesses are thoroughly addressed.


**Justification For Why Not Lower Score:**

It is worth giving the paper a chance due to its solid theoretical justification.


**Metareview: Summary, Strengths And Weaknesses:**

Although there are mixed views and divided opinions among us about this paper, it stands out as a rare contribution to the topic of time series data augmentation by presenting a principled approach with solid theoretical justification. What’s more, the theoretical results are validated experimentally through extensive empirical studies. While it is worth giving this paper a chance to get published, its impact could be higher if the major weaknesses as pointed out by the reviewers can be addressed as much as possible before publication. The authors are highly recommended to revise their paper thoroughly by considering our comments and suggestions.


**Note From Pc:**

if the above contains the word "oral" or "spotlight" please see: "oral" presentation means -> notable-top-5% and "spotlight" means -> notable-top-25%. As stated in our emails, we are disassociating presentation type from AC recommendations